# Light-driven transformable optical agent with adaptive functions for boosting cancer surgery outcomes

Ji Qi [1], Chao Chen[2], Xiaoyan Zhang[2], Xianglong Hu[1], Shenglu Ji[2], Ryan T.K. Kwok[1], Jacky W.Y. Lam[1], Dan Ding[2] & Ben Zhong Tang [1,3]

Fluorescence and photoacoustic imaging have different advantages in cancer diagnosis; however, combining effects in one agent normally requires a trade-off as the mechanisms interfere. Here, based on rational molecular design, we introduce a smart organic nanoparticle whose absorbed excitation energy can be photo-switched to the pathway of thermal deactivation for photoacoustic imaging, or to allow opposed routes for fluorescence imaging and photodynamic therapy. The molecule is made of a dithienylethene (DTE) core with two surrounding 2-(1-(4-(1,2,2-triphenylvinyl)phenyl)ethylidene)malononitrile (TPECM) units (DTE-TPECM). The photosensitive molecule changes from a ring-closed, for photoacoustic imaging, to a ring-opened state for fluorescence and photodynamic effects upon an external light trigger. The nanoparticles' photoacoustic and fluorescence imaging properties demonstrate the advantage of the switch. The use of the nanoparticles improves the outcomes of in vivo cancer surgery using preoperative photoacoustic imaging and intraoperative fluorescent visualization/photodynamic therapy of residual tumours to ensure total tumour removal.

[1] Department of Chemistry, Hong Kong Branch of Chinese National Engineering Research Center for Tissue Restoration and Reconstruction, Division of Life Science, State Key Laboratory of Molecular Neuroscience, Institute for Advanced Study, Institute of Molecular Functional Materials, The Hong Kong University of Science and Technology, Clear Water Bay, Kowloon, Hong Kong, China. [2] State Key Laboratory of Medicinal Chemical Biology, Key Laboratory of Bioactive Materials, Ministry of Education, and College of Life Sciences, Nankai University, Tianjin 300071, China. [3] NSFC Centre for Luminescence from Molecular Aggregates, SCUT-HKUST Joint Research Institute, State Key Laboratory of Luminescent Materials and Devices, South China University of Technology, Guangzhou 510640, China. These authors contributed equally: Ji Qi, Chao Chen. Correspondence and requests for materials should be addressed to D.D. (email: dingd@nankai.edu.cn) or to B.Z.T. (email: tangbenz@ust.hk)

Jablonski diagram helps clarify the basic principles of molecular photophysics, which is closely correlated with the functionality and efficacy of molecular optical agents for cancer diagnosis (e.g. fluorescence and photoacoustic (PA) imaging) and treatment (e.g. photodynamic therapy (PDT))[1–3]. On the basis of Jablonski diagram, there are generally three energy dissipation pathways that probably occur after a chromophore absorbs light[4–6]: (1) fluorescence emission; (2) intersystem crossing to a triplet excited state, followed by generation of phosphorescence and/or reactive oxygen species (ROS) and (3) thermal deactivation via non-radiation pathways. Among them, the absorbed energy for thermal deactivation is usually in direct proportion to the PA effect, as production of heat results in transient thermoelastic expansion and hence ultrasonic waves allowing for PA imaging[7–9]. Since the absorbed excitation energy is fixed in one chromophore, its fluorescent and PA effects are always competitive[10]. It has been well established that quenching the fluorescence of a near-infrared (NIR) light absorbing chromophore is conducive to significantly boosting its PA signal[11, 12]. Therefore, you cannot burn the candle at both ends; that is, utmost fluorescence and PA imaging, never both by far.

However, fluorescence and PA imaging techniques have their own strengths and weaknesses, and more importantly, they have the characteristics of complementary advantages[13]. Fluorescence technique holds the advantage of excellent sensitivity but lacks of spatial resolution[14]. PA technique, on the other hand, offers centimetre-scale deep imaging depth but suffers from low sensitivity[15–17]. Accordingly, the integration of fluorescence and PA imaging modes decidedly enables precise diagnostic outcome by virtue of high sensitivity and imaging depth beyond the optical diffusion limit[18–21]. For this purpose, there have been a number of investigations to date reported that one material with NIR absorption could be simultaneously used for dual-modality fluorescence and PA imaging[20, 21]. Nevertheless, this also implies that such material cannot try its best to do each optimally, as the photophysical working mechanisms of fluorescence and PA are nearly opposite to each other[4, 22, 23]. Therefore, development of an intelligent material with tunable photophysical properties, whose absorbed energy can be controlled to mostly concentrate on either fluorescence or PA channel as needed, is momentously desirable. To our knowledge, unfortunately, no such smart materials have been reported up to present.

In this contribution, we report a smart function-transformable nanoparticle (NP) based on a photo-controllable molecule dithienylethene (DTE)-(1-(4-(1,2,2-triphenylvinyl)phenyl)ethylidene)malononitrile (TPECM) for considerable improvement of cancer surgery outcomes. DTE-TPECM consisting of a DTE core and two surrounding TPECM units has closed-ring and open-ring isomers, reversibly switchable by external UV/visible light irradiation (Fig. 1a). In the ring-closing form, intramolecular energy transfer from TPECM to closed-ring DTE and relatively planar geometric structure make thermal deactivation pathway dominate, leading to utmost absorbed energy focusing on PA imaging. In the ring-opening form, however, both the molecular geometry and photophysical property totally change to make every effort to block the thermal deactivation, hence activating fluorescence emission and ROS production. It is found that the ring-closing NPs generate noticeable PA signal output and possess good signal stabilities, which are superior to several commonly used PA contrast agents including semiconducting polymer nanoparticles (SPNs), methylene blue (MB) and indocyanine green (ICG). Further surface modification of the NPs with a targeting moiety endows them specific tumour-targeting ability. In vivo studies demonstrate that such intelligent NPs with controlled photophysical processes significantly boost the cancer surgery outcomes by harnessing the respective advantages of PA imaging, fluorescence imaging and PDT. This study thus provides a concept of function-transformable optical agent with maximized effectiveness of each function, and verifies its great clinical potential in cancer diagnosis and treatment during surgery.

## Results

**Synthesis and characterization of photo-controllable molecules.** Key synthesis steps of DTE-TPECM are presented in Fig. 1a. Suzuki cross-coupling reaction was carried out between 1-(4-(1,2-diphenyl-2-(4-(4,4,5,5-tetramethyl-1,3,2-dioxaborolan-2-yl)phenyl)vinyl)phenyl)ethan-1-one (**1**) and 3,3′-(perfluorocyclopent-1-ene-1,2-diyl)bis(5-bromo-2-methylthiophene) (**2**) to produce the diketone compound (**3**), which was further reacted with malononitrile to afford ROpen-DTE-TPECM as a yellow powder in a high yield. Detailed synthesis and characterization of the intermediates and final compound with nuclear magnetic resonance (NMR) and high-resolution mass spectrum (HRMS) are shown in Supplementary Methods and Supplementary Figs 1-17. ROpen-DTE-TPECM in THF shows intense absorption below 500 nm (Supplementary Fig. 18). Upon irradiation of such THF solution using 365 nm light for 5 min, ROpen-DTE-TPECM transforms to its ring-closing isomer (RClosed-DTE-TPECM) as evidenced by the occurrence of a new absorption band from 520 to 800 nm (Supplementary Fig. 19a). Noteworthy, DTE-TPECM molecule reversibly switches between the ring-opening and ring-closing states by external UV/visible light exposure (Supplementary Figs 19b,c).

The fluorescence properties of ROpen-DTE-TPECM and RClosed-DTE-TPECM were investigated. ROpen-DTE-TPECM exhibits typical aggregation-induced emission (AIE) feature (Fig. 1b, c): ROpen-DTE-TPECM in good solvent THF is non-emissive due to the low-frequency rotations of surrounding phenyl rings leading to rapid decay of the excited states; however, after formation of ROpen-DTE-TPECM aggregation by adding water (poor solvent) into THF solution, such intramolecular rotations are restricted by intermolecular steric hindrance, resulting in opening the radiative pathway[24]. In marked comparison with the bright fluorescence of ROpen-DTE-TPECM in aggregated state, there is no detectable photoluminescence (PL) emission of the ring-closing isomer in both THF solution and aggregation form even extending the wavelength to 1200 nm (Supplementary Fig. 20), because of the intramolecular energy transfer from the fluorescent TPECM to the non-emissive ring-closing DTE core[25, 26].

Density functional theory gives the optimized geometric structures of RClosed-DTE-TPECM and ROpen-DTE-TPECM (for Cartesian coordinates see Supplementary Tables 1,2). Owing to the closed ring, the two thiophene rings in RClosed-DTE-TPECM form a very planar conjugated structure. Compared with RClosed-DTE-TPECM, ROpen-DTE-TPECM has a more twisted 3D geometry with severely distorted structures of both the DTE core and TPECM arms thanks to the open ring (Fig. 1a), which undoubtedly hinders the intermolecular interactions (e.g. π–π stacking) when aggregated and thus significantly suppresses the non-radiative decay pathways[27, 28]. This hence explains why we choose TPECM as the arms to endow ROpen-DTE-TPECM with AIE effect, i.e. making every effort to block the thermal deactivation. Figure 1d displays the X-ray diffraction (XRD) profiles of the ring-opening and ring-closing isomers. Rather strong diffraction peaks (100, 200, 300 and 010) are observed in RClosed-DTE-TPECM, whereas there is only one weak peak (100) for ROpen-DTE-TPECM. This result verifies that the ring-closing molecules with more planar structure induce much stronger intermolecular interactions[29], agreeing well with the molecular geometry.

**Design principle of photo-controllable molecules.** The molecular design rationale is summarized as follows. For RClosed-DTE-TPECM, the closed ring imparts a long-wavelength

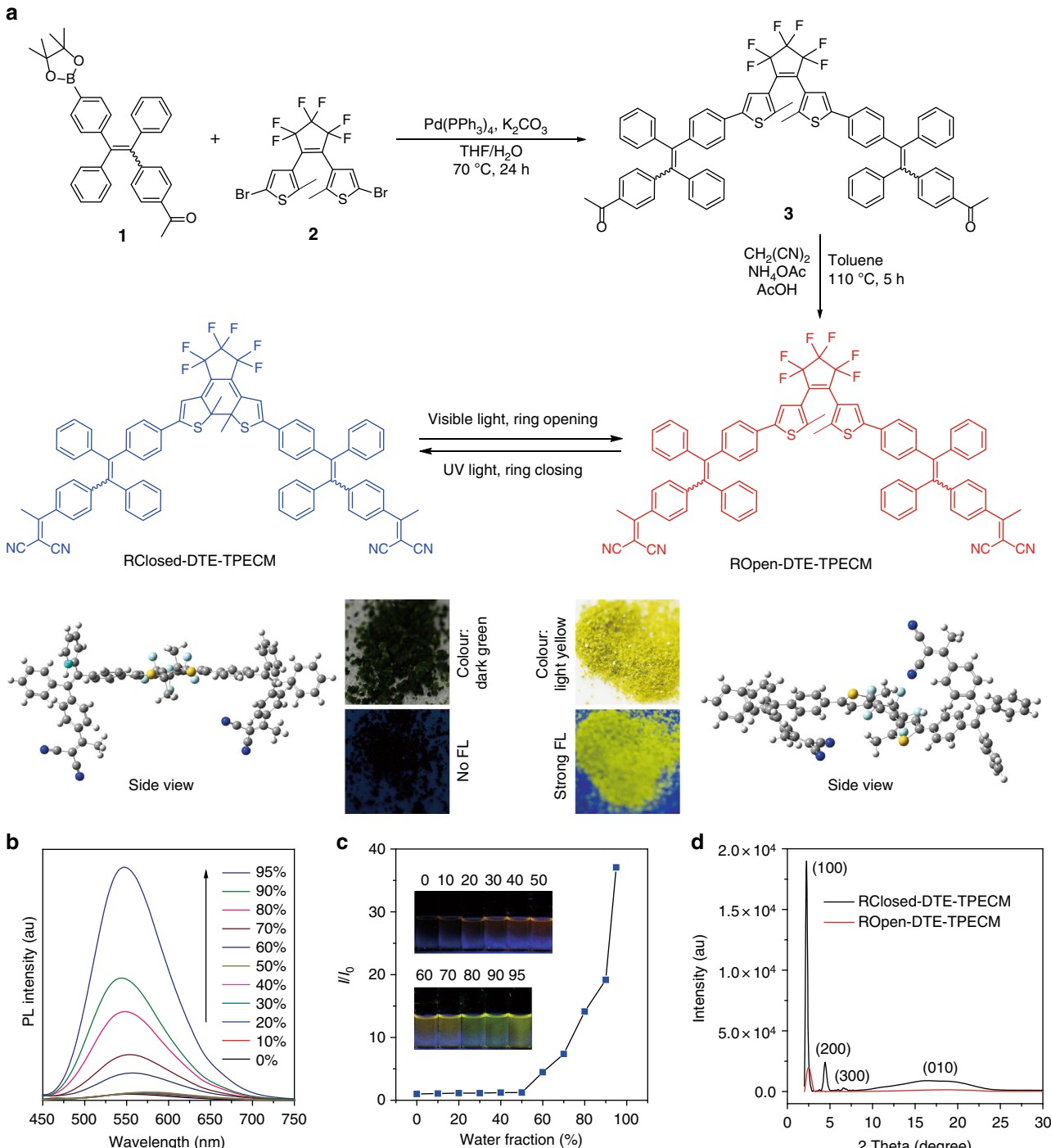

**Fig. 1** Synthesis, structure and property of photo-controllable DTE-TPECM molecules. **a** Key synthesis steps, photo-controlled reversibility and optimized geometric structures of DTE-TPECM molecules. Photographs of ROpen-DTE-TPECM and RClosed-DTE-TPECM powders in daylight and under UV light (365 nm). FL: fluorescence. **b** PL spectra of ROpen-DTE-TPECM in THF/water mixture with various water fractions. **c** Plot of $I/I_0$ versus water fraction. $I_0$ and $I$ are the peak PL intensities of ROpen-DTE-TPECM (10 μM) in pure THF and THF/water mixtures, respectively. Inset shows the photographs of ROpen-DTE-TPECM in THF/water mixtures with different water fractions taken under UV illumination. **d** XRD diagrams of ROpen-DTE-TPECM and RClosed-DTE-TPECM

absorption peak owing to the formation of low-bandgap conjugated structure by the fused dithienylethene. Besides, TPECM is designed to contain tetraphenylethene (TPE) unit and electron-deficient moiety (i.e. malononitrile), forming a donor–acceptor (D–A) structure, which enables efficient intramolecular charge transfer. Incorporation of such D–A fluorophores thus causes a bathochromic shift, making the absorption maximum of

RClosed-DTE-TPECM well match the NIR pulsed laser excitation of PA imaging system. More importantly, in ring-closing state, intramolecular energy transfer occurs that tremendously quenches the fluorescence, and the relatively planar geometric structure of RClosed-DTE-TPECM also promotes intermolecular interactions. These vitally boost the non-radiation pathways[30]. Accordingly, the energy balance of photophysics profoundly tilts

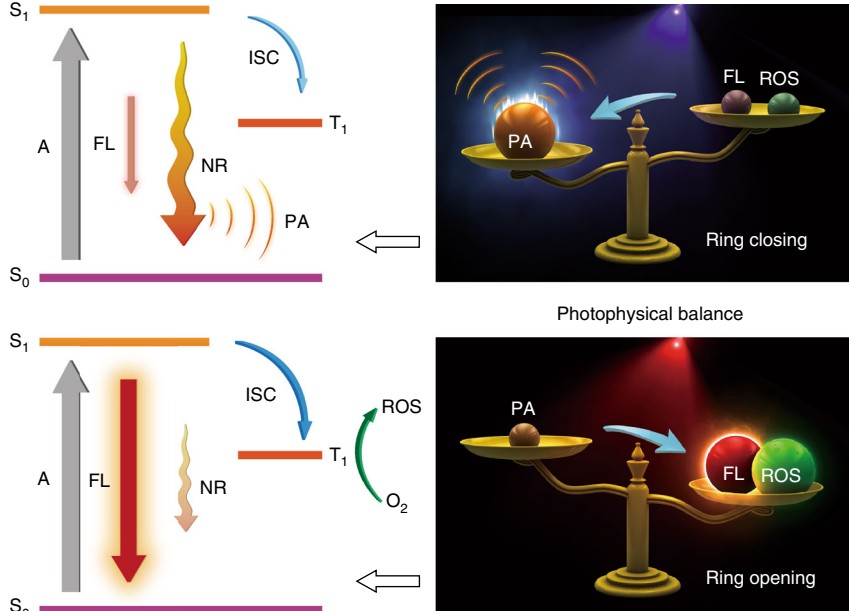

**Fig. 2** Illustration of the controllable photophysical processes. The energy balance of photophysics profoundly tilts to either side controlled by external UV/visible light exposure. A: absorption, FL: fluorescence, NR: non-radiation, ISC: intersystem crossing

to the thermal deactivation side when the ring is closed, benefitting PA transition process (Fig. 2).

On the other hand, upon a simple visible light irradiation, the ring opens to yield ROpen-DTE-TPECM, which not only disrupts intramolecular energy transfer, but also transforms to a much more twisted 3D geometric structure favouring reduced intermolecular interactions. These tremendously inhibit the absorbed energy from flowing to thermal deactivation, and thus adjust the energy balance of photophysics to incline to the opposite side (Fig. 2), i.e. fluorescence emission and ROS generation (as ROpen-DTE-TPECM is not phosphorescent), allowing for fluorescence imaging and PDT. We hypothesized that, simply driven by external light, an overall majority of absorbed energy can be controlled to focus on either side of the balance, which makes our molecule exert its maximum potential for either PA imaging or fluorescence imaging plus PDT, and serve as a powerful optical agent for each different application.

**Preparation and characterization of function-transformable NPs**. To render the hydrophobic organic compounds with good in vivo biocompatibility, a nanoprecipitation method was adopted to formulate RClosed-DTE-TPECM or ROpen-DTE-TPECM using amphiphilic maleimide-bearing lipid-PEG$_{2000}$ as the doping matrix, yielding RClosed-DTE-TPECM-doped or ROpen-DTE-TPECM-doped lipid-PEG$_{2000}$ NPs (in short, RClosed NPs and ROpen NPs, respectively). During the NP formation, the hydrophobic compounds and lipids entangle with each other and their formed aggregates act as the NP core, which is surrounded by the hydrophilic PEG outer layer that stabilizes the NPs (Fig. 3a). Dynamic light scattering (DLS) and transmission electron microscopy (TEM) data show that both RClosed and ROpen NPs are spherical in shape with a similar average diameter of ~65 nm (Fig. 3b, c). As presented in Fig. 3d, RClosed NPs appear in blue-green colour in aqueous solution, which possess an intense absorption peak centred at 650 nm with a molar extinction coefficient of $4.4 \times 10^4\ M^{-1}\ cm^{-1}$ (Supplementary Fig. 21). The PL spectra reveal that RClosed NPs are almost non-fluorescent in water (Fig. 3e and Supplementary Fig. 20). Under continuous visible light (e.g. 610 nm light) irradiation for 10 min, the

absorption peak ranging from 520 to 800 nm gradually decreases and finally vanishes, which is accompanied by the solution colour changed to yellow and the emission peak at ~550 nm significantly intensified, indicating the transformation from RClosed NPs to fluorescent ROpen NPs (Fig. 3d, e). The ring-closing and ring-opening NPs can convert reversibly by alternating UV/visible light irradiation with negligible interference on the absorption, emission and PA properties during ten circles (Fig. 3f and Supplementary Fig. 22), suggesting the highly reversible and bistable photochromism signature. It is also found that the RClosed NPs can effectively change to ROpen NPs even if the 610 nm red light irradiation ($0.3\ W\ cm^{-2}$) is through a 1 cm thickness of chicken breast (Supplementary Fig. 23).

**PA property of the ring-closing NPs**. The PA properties of RClosed NPs and ROpen NPs were studied by recording the PA intensity at different wavelengths from 680 to 840 nm. RClosed NPs effectively generate PA signals under NIR pulsed laser irradiation with PA spectrum in good accordance with the absorption profile in the NIR region, whereas there is negligible PA signal detected from ROpen NPs (Fig. 4a). A linear relationship is observed between PA intensity at 700 nm and NP concentration based on RClosed-DTE-TPECM (Fig. 4b). It is worthy to note that after exposure to $1.8 \times 10^4$ laser pulses at 700 nm ($1.5\ W\ cm^{-2}$ laser and 20 Hz pulse repetition rate), nearly no loss of PA intensity is observed for RClosed NPs (Fig. 4c), revealing the good photostability of RClosed NPs, which hardly convert to ROpen NPs under the PA imaging condition at 700 nm.

The PA signal and stability of RClosed NPs were then compared with several popularly used PA contrast agents, including SPNs, MB and ICG. The SPNs were prepared according to the literature by formulation of a semiconducting polymer poly(cyclopentadithiophene-*alt*-benzothiadiazole) using lipid-PEG$_{2000}$ as the encapsulation matrix (Supplementary Fig. 24)[4, 31]. The absorption spectra of SPNs, MB and RClosed NPs in water suggest that they share similar maximal absorption wavelength (Supplementary Fig. 25). Moreover, as the SPNs, MB and RClosed NPs are spectrally similar with PA maximum wavelength at about 680 nm[31, 32], rational comparison is allowed

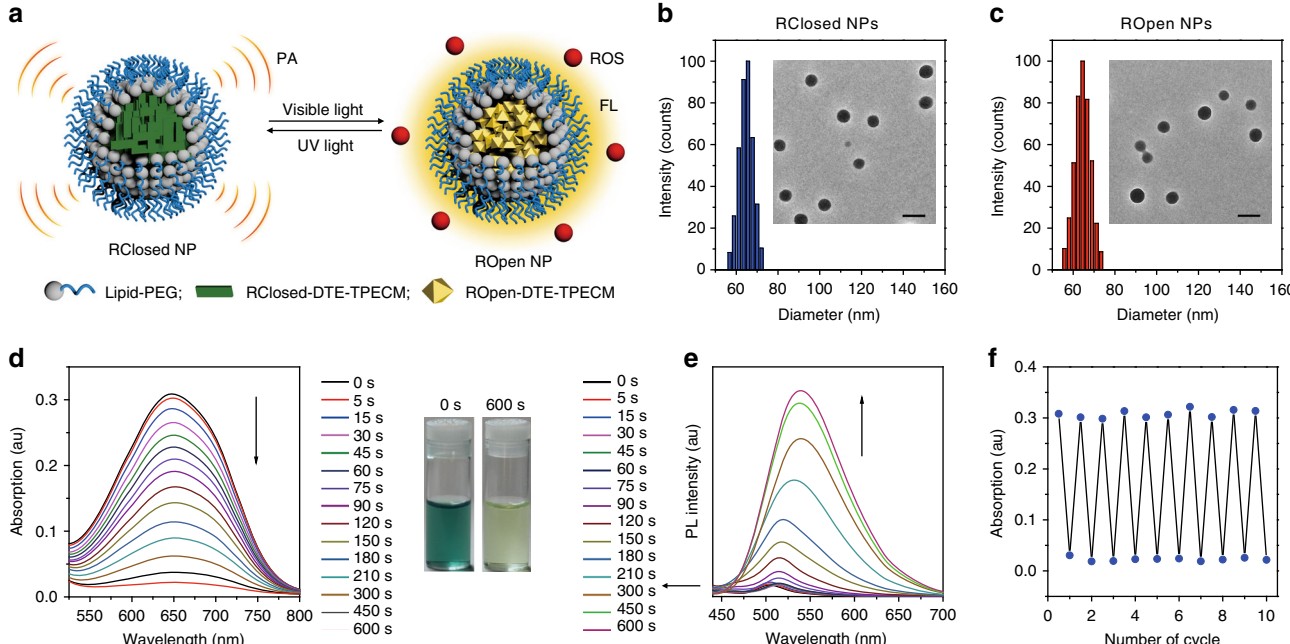

**Fig. 3** Preparation and characterization of the photo-controllable NPs. **a** Schematic of RClosed NPs and ROpen NPs. **b**, **c** DLS profiles and TEM images of **b** RClosed NPs and **c** ROpen NPs. Scale bars, 100 nm for TEM images. **d** Absorption and **e** PL spectra of RClosed NPs under visible light (610 nm, 0.3 W cm$^{-2}$) irradiation for different time as indicated. Photographs in **d** indicate the aqueous solutions of RClosed NPs before and after 610 nm light exposure for 600 s. **f** The absorption intensity at 650 nm of the NPs during ten circles of visible (610 nm, 0.3 W cm$^{-2}$)/UV light (365 nm, 0.1 W cm$^{-2}$) irradiation processes

using a 680 nm pulsed laser. At the same condition, the PA intensity of RClosed NPs is ~1.8-fold and ~2.0-fold higher than that of SPNs and MB, respectively (Fig. 4d). Since the SPNs have been demonstrated as a high-performing PA contrast agent even superior to single-walled carbon nanotubes[31] and MB is also a commonly used molecule for PA imaging[16], this comparison result illustrates that RClosed NPs can serve as an advanced PA molecular probe. High stability of probe signal against tumour endogenous reactive oxygen and nitrogen species (RONS) such as hypochlorite (ClO$^-$), peroxynitrite (ONOO$^-$) and hydroxyl radical (•OH) is an important prerequisite for accurate cancer diagnosis[33–35]. As depicted in Fig. 4e, in the presence of various RONS, the absorption spectra of RClosed NPs and SPNs hardly change, indicating that they are RONS-inert. In sharp contrast, ICG shows the worst performance in resisting RONS with absorption band rapidly decreasing after addition of each RONS. Moreover, MB is not stable against ONOO$^-$, regardless of the good resistance towards ClO$^-$ and •OH. As SPNs are advantageous due to their excellent RONS resistance[36, 37], the result proves that RClosed NPs are promising for precise in vivo PA cancer imaging.

**Fluorescence property and ROS generation of the ring-opening NPs.** PL excitation mapping was performed on ROpen NPs, displaying excitation and emission peaked at ~410 and ~550 nm, respectively (Fig. 4f). The NIR absorption completely disappears after RClosed NPs transform to ROpen NPs. The fluorescence quantum yield ($\Phi_F$) and lifetime ($\tau$) of ROpen NPs are measured to be 23.8% and 1.45 ns (Fig. 4g), respectively, while RClosed NPs show no detectable $\Phi_F$ and $\tau$. The ROpen NPs are tolerant to ClO$^-$, ONOO$^-$ and •OH, as indicative of the unchanged emission spectrum in the presence of each RONS (Supplementary Fig. 26). The ROpen NPs also exhibit similar anti-photobleaching capacity to commercial QD585 (Supplementary Fig. 27), which is well-known for its ultrahigh photobleaching threshold[38].

For photosensitizers, the absorbed energy can transfer to the triplet excited state via intersystem crossing, followed by generation

of ROS that are singlet oxygen in most cases[5, 39–41]. The abilities of the RClosed and ROpen NPs to produce ROS upon light excitation were examined utilizing 2′,7′-dichlorodihydrofluorescein diacetate (DCF-DA) as the ROS indicator[42]. As depicted in Fig. 4h, RClosed NPs hardly generate ROS upon excitation at either 365 nm or 610 nm. On the contrary, efficient ROS generation is observed for ROpen NPs upon excitation at 365 nm by monitoring the fluorescence activation due to the oxidation reaction between ROS and non-emissive DCF-DA to yield fluorescent dichlorofluorescein (DCF)[42]. Additionally, after conversion of RClosed NPs to ROpen NPs via exposure to 610 nm light, the converted ROpen NPs can efficiently produce ROS under subsequent 365 nm light irradiation. It is also validated that ROpen NPs are capable of effectively generating ROS upon exposure to white light (Fig. 4i), as white light (400–700 nm) irradiation has been widely accepted for in vivo PDT[43].

**NP surface modification with a targeting ligand.** It has been reported that the peptide with a sequence of YSAYPDSVPMMS (named YSA in short) is able to selectively and tightly bind to EphA2 protein, which is a transmembrane receptor tyrosine kinase overexpressed in many cancer cells as well as tumour blood vessels including 4T1 mammary adenocarcinoma[44, 45]. Therefore, YSA peptide was employed as a targeting ligand to modify our function-transformable NPs in order to endow them with active tumour-targeting ability. We synthesized CYSAYPDSVPMMS peptide (Fig. 5a) with a terminal thiol group in cysteine (C) via standard 9-fluorenylmethoxycarbonyl (Fmoc) solid-phase peptide synthesis (SPPS), which was characterized by LC–MS and HRMS (Supplementary Figs 28,29, and details see Supplementary Methods). Our function-transformable NPs were then modified with CYSAYPDSVPMMS peptide through the coupling reaction between the thiol group of peptide and the maleimide group of PEG on the NPs, affording YSA-conjugated NPs (namely RClosed-YSA NPs or ROpen-YSA NPs; Fig. 5b). It is calculated that there are ~3700 YSA peptides on average

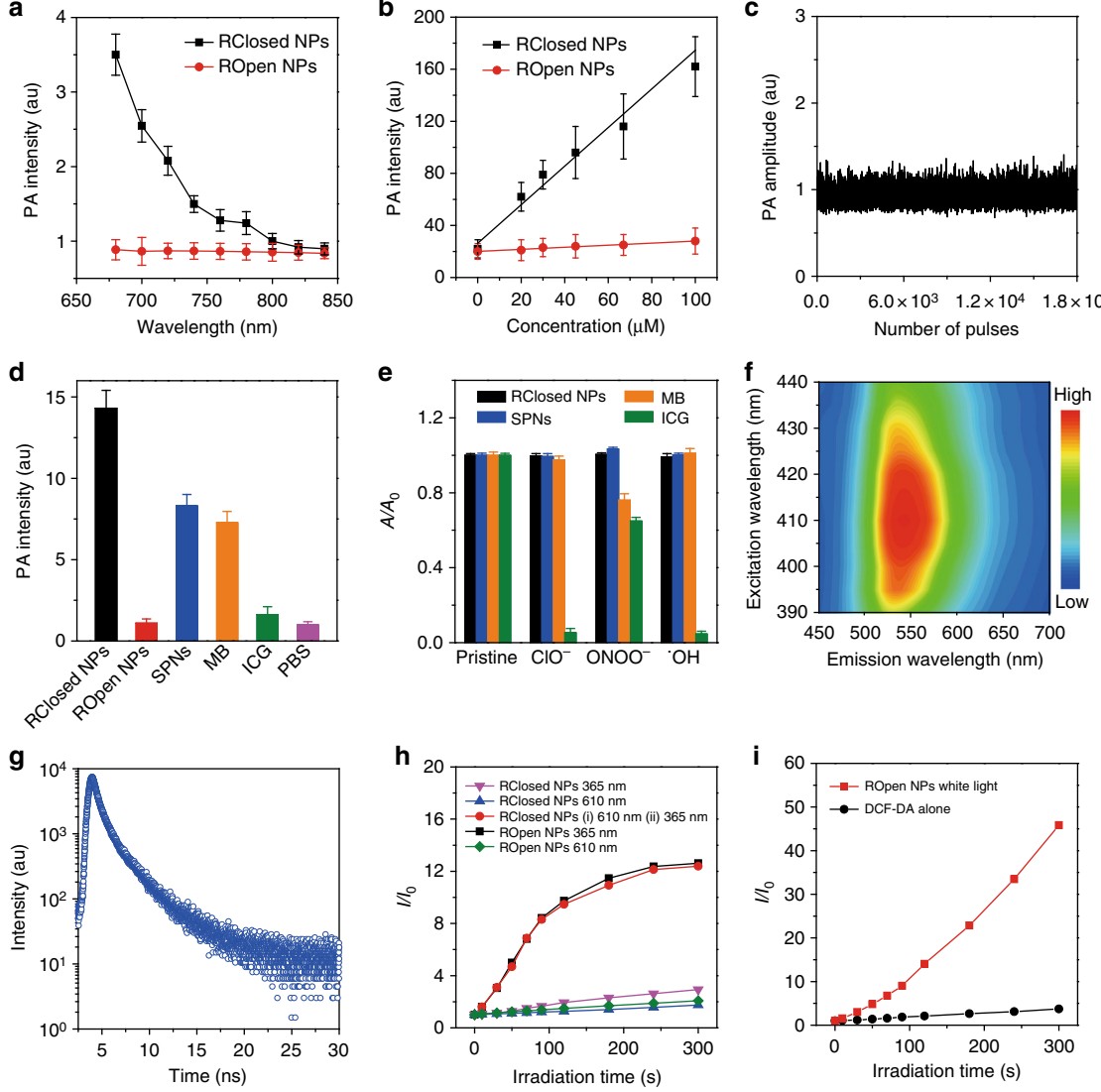

**Fig. 4** In vitro PA, fluorescence and ROS generation of the NPs. **a** PA spectra of RClosed and ROpen NPs. **b** PA intensities of RClosed and ROpen NPs at 700 nm as a function of molar concentration based on DTE-TPECM molecules. Error bars, mean ± s.d. ($n = 3$) for **a**, **b**. **c** PA amplitudes of RClosed NPs as a function of number of laser pulses ($1.8 \times 10^4$ pulses; 1.5 W cm$^{-2}$ laser and 20 Hz pulse repetition rate). **d** PA intensities excited with 680 nm pulsed laser of various agents at the same molar concentration (100 μM) based on MB, ICG, DTE-TPECM molecules and the repeat unit of SP. **e** Plot of $A/A_0$ versus different RONS. $A$ and $A_0$ are the absorption intensity at 680 nm of RClosed NPs, SPNs, MB and ICG in the presence and absence of RONS (400 μM), respectively. Error bars, mean ± s.d. ($n = 3$) for (**d**, **e**). **f** PL excitation mapping and **g** fluorescence decay curve of ROpen NPs. **h** Plot of $I/I_0$ versus light irradiation time. The aqueous solution of RClosed NPs or ROpen NPs (10 μM based on DTE-TPECM) was exposed to 610 nm red light (0.3 W cm$^{-2}$) and/or 365 nm UV light (0.1 W cm$^{-2}$). **i** Plot of $I/I_0$ versus white light (0.25 W cm$^{-2}$) irradiation time of ROpen NPs (10 μM based on ROpen-DTE-TPECM) in aqueous solution. $I_0$ and $I$ are the PL intensity of DCF at 525 nm before and after light irradiation at designated time intervals in both **h**, **i**

conjugated on each NP (Supplementary Methods Equation 3). The mean size of YSA-conjugated NPs is ~68 nm determined by DLS, which is similar to that of the NPs without YSA. It is also demonstrated that the YSA modification does not influence any of the NP properties in terms of PA and fluorescence properties, ROS generation capacity and reversible photochromism.

In vitro cellular study was then carried out with 4T1 murine breast cancer cells. Using hepatic L02 normal cells as a control, the western blot study reveals that EphA2 is predominantly expressed in 4T1 cancer cells (Fig. 5c and Supplementary Fig. 30). It is demonstrated that the YSA conjugation significantly improves the NP internalization by 4T1 cancer cells because of the strong interaction between YSA and EphA2 receptor overexpressed on the cancer cell membrane[44, 45], and that the RClosed-YSA NPs can be facilely transformed to the ROpen-YSA

NPs inside the 4T1 cells, triggered by 610 nm light irradiation (Supplementary Figs 31 and 32). Additionally, 610 nm light (0.3 W cm$^{-2}$) irradiation itself does not cause the photothermal effect (Supplementary Fig. 33) and the treatment of 'RClosed-YSA NPs + 610 nm light irradiation (0.3 W cm$^{-2}$, 5 min)' results in negligible cytotoxicity (Fig. 5d), implying the good biocompatibility of the NPs and the harmless of red light irradiation under the experimental condition. It is also found that the converted ROpen-YSA NPs within 4T1 cancer cells have good ROS generation ability (Supplementary Fig. 34) and more effective in vitro PDT efficacy than the converted ROpen NPs (Fig. 5e).

**In vivo pharmacokinetics and biodistribution.** After we demonstrated that both the RClosed-YSA and ROpen-YSA NPs

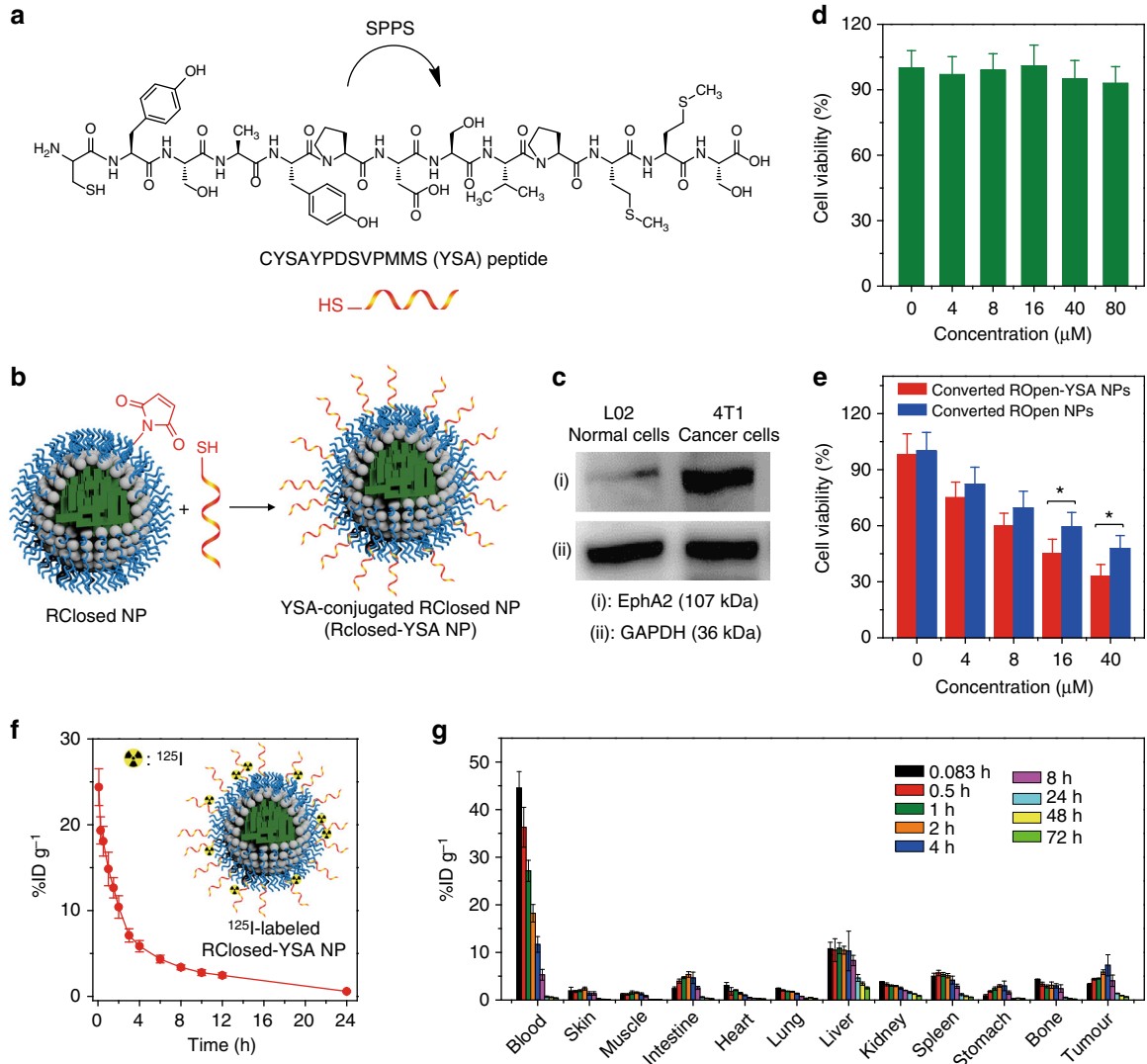

**Fig. 5** Targeting modification, cytotoxicity and in vivo pharmacokinetics. **a** Chemical structure of CYSAYPDSVPMMS peptide. SPPS solid-phase peptide synthesis. **b** Schematic of the preparation of a RClosed-YSA NP. **c** Western blot analyses of EphA2 in L02 cells and 4T1 cancer cells. **d** Cell viability of RClosed-YSA NP-incubated 4T1 cancer cells after 610 nm red light (0.3 W cm$^{-2}$) irradiation for 5 min. Error bars, mean ± s.d. ($n = 4$). **e** Cell viabilities of the converted ROpen-YSA NP-loaded and converted ROpen NP-loaded 4T1 cancer cells under white light irradiation (0.25 W cm$^{-2}$, 4 min). Error bars, mean ± s.d. ($n = 4$). *$P < 0.05$, unpaired Student's $t$-test (two-tailed). The cells were incubated with RClosed-YSA and RClosed NPs, respectively, followed by exposure to 610 nm red light (0.3 W cm$^{-2}$) for 5 min, to obtain the converted ring-opening NP-loaded cells. In **d** and **e**, the concentration is based on DTE-TPECM. **f** Pharmacokinetics study of $^{125}$I-labelled RClosed-YSA NPs analysed by scintillation count of $^{125}$I radioactivity in blood. Error bars, mean ± s.d. ($n = 6$ rats). Inset displays the schematic of an $^{125}$I-labelled RClosed-YSA NP. **g** Biodistribution of $^{125}$I-labelled RClosed-YSA NPs in various tissues of 4T1 tumour-bearing mice at different time points post-intravenous injection. Error bars, mean ± s.d. ($n = 6$ mice for each time point)

can be safely utilized for in vivo application through a series of blood chemistry examinations and histological analyses of important normal organs (Supplementary Figs 35-37), in vivo pharmacokinetics of the function-transformable NPs was investigated. As radiolabelling is a routine and reliable method to trace administered species in in vivo pharmacokinetic studies[46], RClosed-YSA NPs were radiolabelled with a radioactive nuclide, iodine-125 ($^{125}$I), which was reacted with the tyrosine (Y) residues of YSA peptide. The radiochemical purity of $^{125}$I-labelled RClosed-YSA NPs is higher than 99%, which does not change upon keeping the NPs in saline for 3 days, revealing the high radiolabelling stability. As RClosed-YSA NPs serve as the staple imaging probe/therapeutic agent in the next cancer surgery study, their pharmacokinetics was evaluated in healthy rats benefitting from $^{125}$I labelling. After $^{125}$I-labelled RClosed-YSA NPs were intravenously injected into the rats, the blood samples were collected at designated

time intervals and counted for $^{125}$I radioactivity with a gamma counter. Figure 5f displays the blood circulation behaviour of $^{125}$I-labelled RClosed-YSA NPs. The circulation half-life, the volume of distribution and the blood clearance of the RClosed-YSA NPs are determined to be 6.21 ± 0.39 h, 143.03 ± 12.12 mL kg$^{-1}$ and 24.47 ± 2.38 mL kg$^{-1}$ h$^{-1}$, respectively (see Supplementary Table 3 for complete pharmacokinetic data).

The biodistribution of $^{125}$I-labelled RClosed-YSA NPs in tumour-bearing mice was investigated as well. The xenograft 4T1 tumour-bearing mouse model was employed, which was established by subcutaneous inoculation of 4T1 cancer cells into the mouse right axillary space. After $^{125}$I-labelled RClosed-YSA NPs were administrated into 4T1 tumour-bearing mice through the tail vein, the time-dependent biodistributions of the NPs in blood, tumour and various major organs of mice were quantitatively analysed by gamma scintillation counting (Fig. 5g).

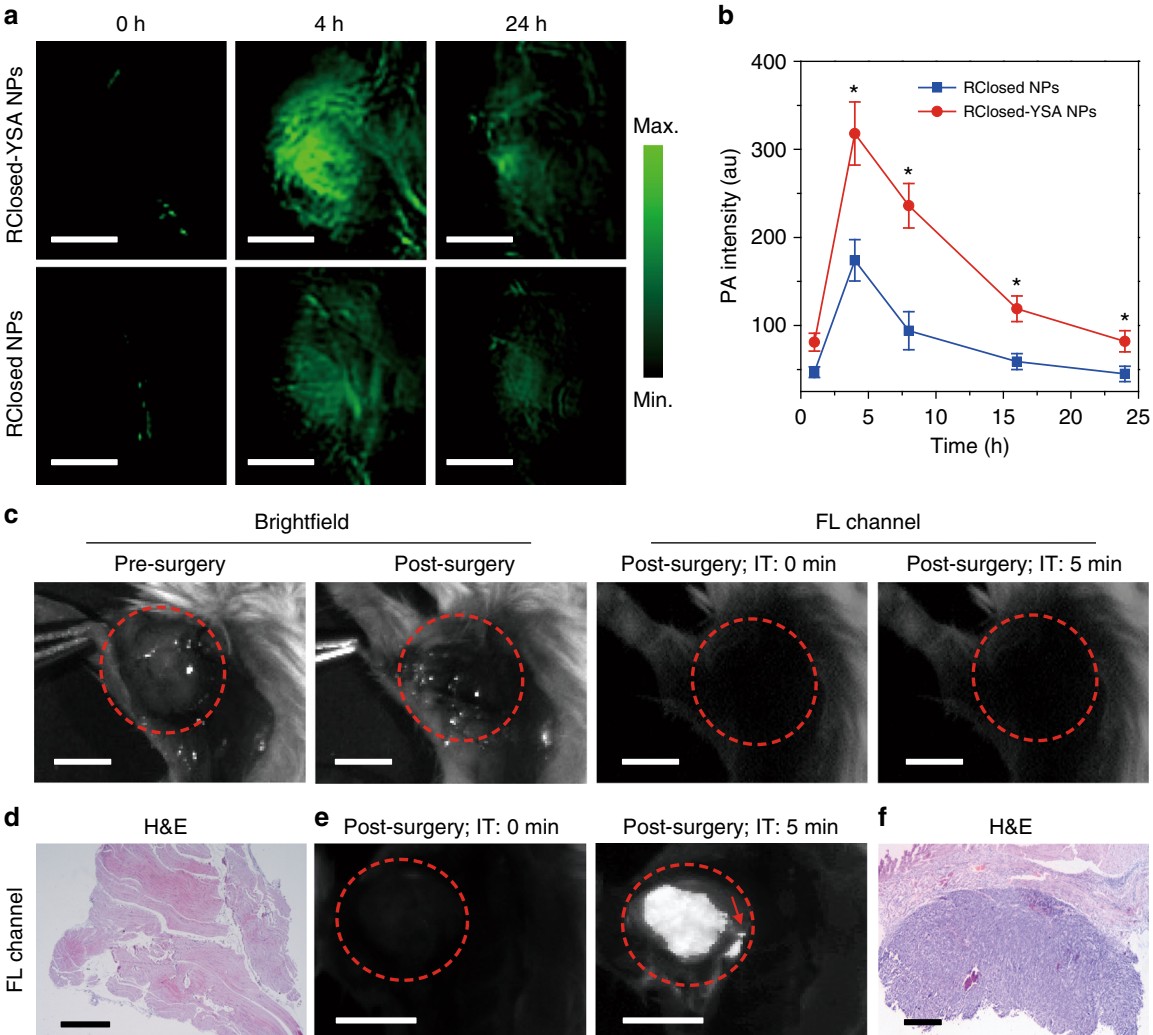

**Fig. 6** In vivo preoperative PA imaging and intraoperative fluorescence imaging. **a** Representative time-dependent PA images of subcutaneous tumours from mice intravenously injected with RClosed-YSA and RClosed NPs (800 μM based on RClosed-DTE-TPECM, 100 μL), respectively. Scale bars, 2 mm. **b** Plot of PA intensity at 700 nm in tumour versus time post injection of RClosed-YSA or RClosed NPs. Error bars, mean ± s.d. ($n = 3$ mice per group). *$P < 0.05$, in comparison between RClosed-YSA and RClosed NPs using unpaired Student's $t$-test (two-tailed). **c** Representative brightfield images of RClosed-YSA NP-treated tumour-bearing mice before and after surgery as well as representative fluorescence images of mice with complete surgical resection of tumours, followed by 610 nm red light (0.3 W cm$^{-2}$) irradiation at the operative incision site for 5 min. Scale bars, 3 mm. FL fluorescence, IT irradiation time. **d** H&E stained tissues at the operative incision site in **c** indicate no residual tumours left behind. Scale bar, 1 mm. **e** Representative fluorescence images of RClosed-YSA NP-treated mice with residual tumours post surgery. The operative incision site was irradiated by 610 nm red light (0.3 W cm$^{-2}$) for 5 min. The red dashed circles in **c** and **e** indicate the tumour/operative incision site. The red arrow shows the residual tumours with a diameter below 1 mm. Scale bars, 3 mm. **f**, H&E stained tissues at the operative incision site in **e** confirm the existence of residual tumours. Scale bar, 0.5 mm

It is obvious that the RClosed-YSA NPs rapidly leave the bloodstream and enter most organs, and the levels of the NPs in all of the tissues significantly decrease after 8 h. Due to the reticuloendothelial system and mononuclear phagocyte system uptake[47], high accumulations of the NPs in liver, spleen and bone marrow are found. Importantly, thanks to both the active (YSA–EphA2 interaction) and passive (the enhanced permeability and retention (EPR) effect of nanomaterials)[48] tumour-targeting capabilities, the RClosed-YSA NPs can be largely enriched in tumour tissue with maximum tumour uptake of ~7.3% ID g$^{-1}$ occurring at 4 h post injection.

**Improvement of cancer surgery outcomes**. We next investigated whether the function-transformable NPs could improve cancer surgery outcomes. As PA technique permits imaging that

surpasses the limit of optical diffusion[15, 16], compared with fluorescence imaging, PA imaging could offer relatively deeper information on the tumours in vivo before surgery. A commercial small-animal opt-acoustic tomography system (MOST) was used to study the utility of RClosed-YSA NPs in in vivo PA imaging of tumours. As shown in Fig. 6a, before NP administration (0 h), there is weak PA signal at 700 nm in the tumours of living mice, probably attributed to the intrinsic background by oxyhemoglobin and deoxyhemoglobin[4]. Subsequently, the RClosed-YSA NPs (100 μL, 800 μM based on RClosed-DTE-TPECM) were injected into one group of 4T1 tumour-bearing mice via the tail vein. As a control, the same amount of RClosed NPs without YSA modification was intravenously administrated into the other group of tumour-bearing mice. In vivo PA imaging was then conducted after injections. For the mice in both two groups, the PA signals in tumours significantly elevate and reach the maximum at 4 h,

which then gradually decrease as the time elapses (Fig. 6a, b), agreeing well with the biodistribution data. Noteworthy, the average PA signal in RClosed-YSA NP-treated tumours is statistically higher than that in RClosed NP-treated tumours at each tested time point (for example, ~1.8 times higher at 4 h post injection) (Fig. 6b).

With the information provided by preoperative PA imaging, surgery can be performed to excise the tumours in vivo. In the clinic, one of the most challenging issues during cancer surgery is to quickly assess whether all the tumour masses have been removed without any residual tumours left behind[49]. Addressing this challenge requires a highly sensitive imaging modality in combination with a highly effective contrast agent. In this regard, fluorescence imaging is a promising candidate, since it is sensitive, fast, real-time and instrument portable[50]. In our experiment, after tumour resection with the aid of PA imaging using RClosed-YSA NPs, 610 nm light was immediately irradiated at the operative incision site for 5 min. Interestingly, if the tumours are totally removed by the surgeon, which is confirmed by hematoxylin and eosin (H&E) histological analyses, no fluorescent signal can be detected at/around the incision site (Fig. 6c, d). Nevertheless, if there are residual tumours left behind post resection, the fluorescent signal gradually turns on at/around the incision site within 5 min irradiation duration (Fig. 6e), which arises from the RClosed-YSA NPs in the residual tumours rapidly transforming to fluorescent ROpen-YSA NPs. The existence of residual tumours is verified by H&E staining (Fig. 6f). As a control, when residual tumours of saline-treated mice are irradiated with 610 nm light for 5 min, no detectable fluorescent signal can be seen (Supplementary Fig. 38). This confirms that the light-up fluorescence from residual tumours indeed originates from the transformed ROpen-YSA NPs. It is important to note that the ratio of average fluorescence intensity from the residual tumours to that from surrounding normal tissues is ~7.1, which outperforms the Rose criterion and is higher than the reported values of ICG and MB in fluorescence imaging-guided surgery[13, 44]. Thanks to the large signal-to-background ratio, submillimeter tumours can also be clearly delineated by the light-up fluorescence of our NPs, indicated by the red arrow in Fig. 6e. For the mice with transformed ROpen-YSA NPs indicating negligible residual tumours, 18 of 20 mice were cured without any in situ tumour recurrences and survived 2 months. On the other hand, all the 20 mice with residual tumours visualized by converted ROpen-YSA NPs fluorescence experienced fast growth of residual tumours and died within 2-month monitoring duration. Thereby, our function-transformable NPs can improve cancer surgery outcomes by preoperative cancer diagnosis via PA imaging together with intraoperative fluorescent visualization of residual tumours in a sensitive, fast and real-time manner, significantly reducing the risk of in situ tumour recurrence.

In many clinical cases, complete tumour resection is impossible or not suggested. Aiming for this, surgical debulking of tumours that refers to removal of most of a surgically incurable malignant tumour has been advocated for many cancers such as ovarian carcinoma, lymphoma, sarcoma and neoplasms of central nervous system, which is very common in the clinic[51]. One major purpose of debulking is to improve the quality of life and extend survival despite of not curing the cancer thoroughly. Generally, subsequent treatment after debulking surgery must be carried out to control the tumours left behind[52]. As the ring-opening NPs serve as an efficient photosensitizer, we wonder whether the PDT of ring-opening NPs within residual tumours post debulking surgery can significantly impede the residual tumour growth and thus prolong the patients' lifetimes. As such, subcutaneous 4T1 xenograft tumour-bearing mice were randomly assigned to five groups, named 'debulking surgery (DS) alone',

'DS + Light', 'DS + YSA NPs', 'DS + NPs + Light' and 'DS + YSA NPs + Light', respectively. It is worthy pointing out that the 4T1 cancer cells in this study express luciferase, allowing for tracking the tumours via bioluminescence imaging (for details see the Methods section). On day 0, RClosed-YSA NPs were intravenously injected into the mice in both 'DS + YSA NPs' and 'DS + YSA NPs + Light' cohorts. Moreover, RClosed NPs were intravenously administrated into the mice in 'DS + NPs + Light' group. At 4 h post injection, the tumours of all the mice in five groups were debulked. The mice sharing similar residual tumours in terms of size and activity determined by bioluminescence imaging were selected with each group containing ten mice. For the mice in 'DS + Light', 'DS + YSA NPs', 'DS + NPs + Light' and 'DS + YSA NPs + Light' groups, after debulking, 610 nm light irradiation (0.3 W cm$^{-2}$) was performed at the incision site for 5 min to convert ring-closing NPs to ring-opening NPs in the residual tumours. This was followed by white light irradiation for another 5 min on residual tumours from mice in 'DS + Light', 'DS + NPs + Light' and 'DS + YSA NPs + Light' groups to make ring-opening NPs generate ROS for PDT.

After various treatments on day 0, the size and activity of residual tumours from mice in all five groups were monitored for 15 days by bioluminescence imaging. As shown in Fig. 7a, b, as compared to DS alone causing fast growth of residual tumours, DS followed by PDT with no matter RClosed-YSA NPs or RClosed NPs gives rise to good efficacy on suppression of residual tumours, as evidenced by the growth stoppage of residual tumours in both 'DS + YSA NPs + Light' and 'DS + NPs + Light' groups. It is worthy to note that 'DS + YSA NPs + Light' is the only treatment that achieves smaller average tumour size on day 15 than that on day 0, leading to better antitumor efficacy as compared to 'DS + NPs + Light'. As controls, the treatments of 'DS + Light' and 'DS + YSA NPs' fail to be efficacious on residual tumour inhibition (Fig. 7b and Supplementary Fig. 39), demonstrating that the impressive antitumour activity indeed roots in the PDT of YSA-conjugated NPs post DS. Furthermore, 9 of 10 mice in 'DS + YSA NPs + Light' cohort and 7 of 10 mice in 'DS + NPs + Light' group could survive 40 days, whereas the mice in the other three groups all died within 40-day study duration (Fig. 7c). These results manifest that our strategy using the function-transformable NPs is also efficacious to improve DS outcomes, effectively prolonging the lifetimes of tumour-bearing mice after DS.

## Discussion

RClosed-DTE-TPECM is rationally designed to make every effort to concentrate utmost absorbed energy on the pathway of thermal deactivation for PA imaging, in terms of intramolecular energy transfer to quench fluorescence, and relatively planar geometric structure to promote intermolecular interactions (Fig. 1a, d). These enable RClosed NPs neither fluoresce nor generate any ROS, but generate brighter PA signal than the reported high-performing SPNs and MB (Fig. 4d)[4, 31]. Upon simple irradiation by visible light, the DTE ring opens to afford ROpen-DTE-TPECM, which however is designed to make every effort to restrain the thermal deactivation pathway, favouring absorbed energy flow to the other two energy dissipation pathways, i.e. fluorescence emission and intersystem crossing to triplet excited state to generate ROS. In addition to open ring cancelling intramolecular energy transfer, AIE-active TPECM is employed to endow the molecule with a clawed 3D geometry for depressing intermolecular interactions (Fig. 1a, d). These significantly block the non-radiative decay, leading to ROpen NPs generating negligible PA signal (Fig. 4a, b) and thus becoming an effective fluorescent probe (Fig. 4f, g) and photosensitizer (Fig. 4h, i).

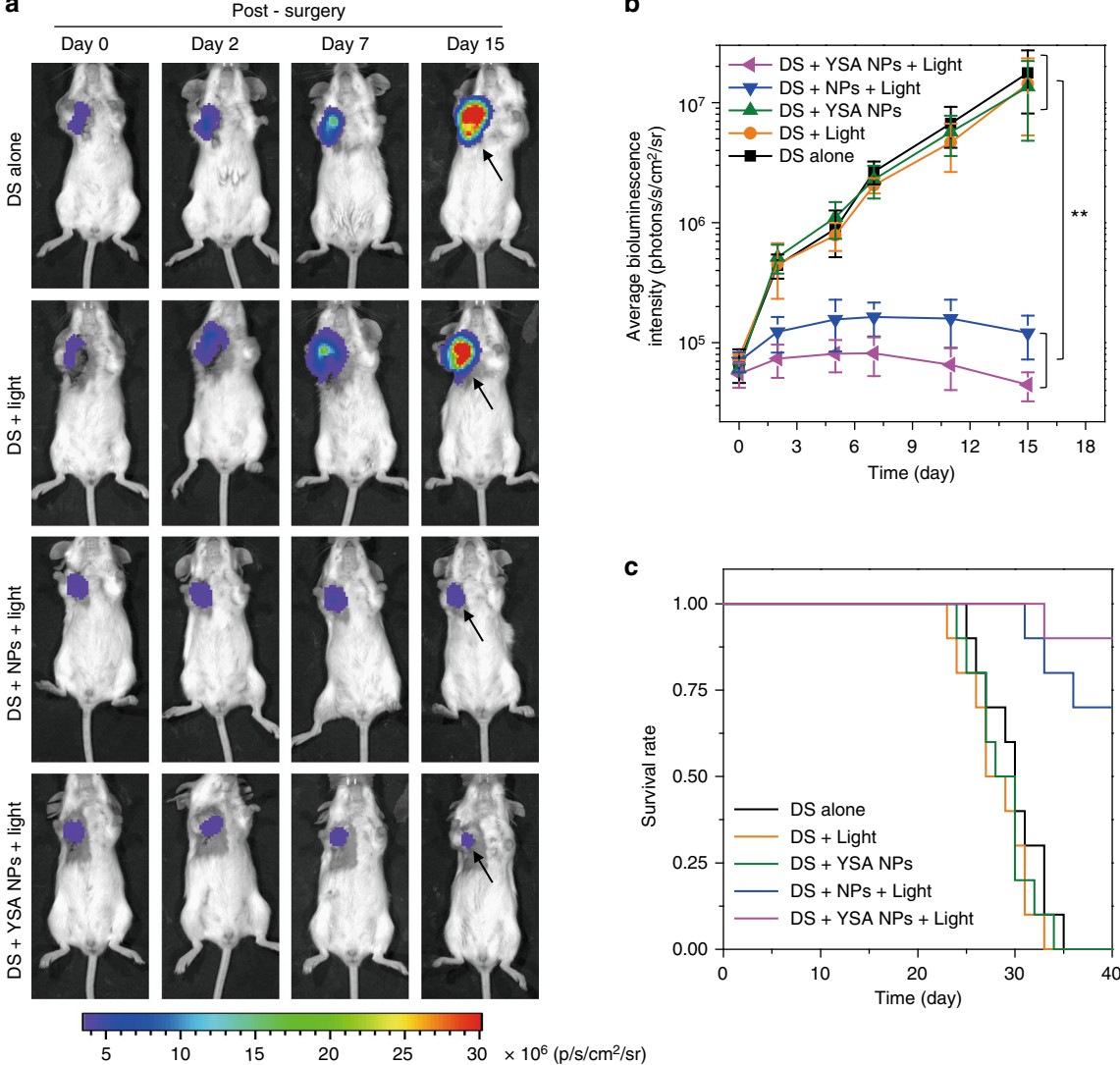

**Fig. 7** In vivo PDT of residual tumours after debulking surgery (DS). **a** Time-dependent bioluminescence imaging of residual tumours from mice in different groups. The tumours were debulked on day 0. The 4T1 cancer cells express luciferase, permitting bioluminescence imaging. The black arrows indicate the residual tumours. **b** Quantitative analysis of bioluminescence intensities of residual tumours from mice with various treatments as indicated. Error bars, mean ± s.d. ($n = 10$ mice per group). **$P < 0.01$, one-way ANOVA. **c** Survival curves for different groups ($n = 10$ mice per group)

Surgery is one of the most adapted strategies to treat solid tumours[52]. For clinical cancer surgery, preoperative imaging and intraoperative imaging call for different imaging techniques[50]. The RClosed-YSA NPs after intravenous administration are capable of delineating tumours via PA imaging before surgery, attributable to their high PA brightness as well as their active (YSA–EphA2 interaction) and passive (EPR effect) tumour-targeting capabilities (Fig. 6a, b). If there are residue tumours post-resection, the transformed fluorescent ROpen-YSA NPs can sensitively visualize them via simple exposure of operative incision site to 610 nm light for 5 min, which give a high tumour-to-normal tissue ratio of ~7.1 and even permit clear detection of residual tumours below 1 mm in diameter (Fig. 6e). As a consequence, our light-driven function-transformable NPs show good performances in both preoperative PA imaging and intraoperative fluorescence imaging, significantly reducing the risk of in situ tumour recurrence.

In cases of DS that complete tumour resection is impossible or not suggested, the PDT of transformed ROpen-YSA NPs within residual tumours post DS are efficacious on suppression of residual tumour growth and prolongation of the lifetimes of tumour-bearing mice (Fig. 7). Such dramatic PDT efficacy not only lies in the effective ROS production of ROpen-YSA NPs, but more importantly, attributes to surgery helping overcome two major limitations of PDT, i.e. limited tissue penetration depth of excitation light and insufficient oxygen within big tumours[53, 54]. Therefore, this study demonstrates that PDT is quite suitable and efficacious for treatment of residual tumours after DS and our smart NPs remarkably promote the DS outcomes.

In summary, we have developed a function-transformable NP that can serve as powerful PA contrast agent, fluorescent probe and photosensitizer as needed, simply triggered by external light, which give excellent performance in boosting the cancer surgery outcomes. Such smart NPs with controlled photophysical properties show unique merits over all other existing optical agents in terms of the combined advantages of simple but 'one-for-all' system, on-demand function tunability and utmost effectiveness of each function. This study therefore creates a class of optical agents with absorbed energy-convertible and function-transformable signatures for advanced biomedical application at a comprehensive level not achievable by currently reported optical

agents. However, the current work still has limitation, as the excitation and emission of the ring-opening NPs are not ideal for practical in vivo applications. Future work will focus on developing function-transformable optical agents with fluorescence imaging in the NIR spectral region.

## Methods

**Preparation of the RClosed NPs and ROpen NPs.** In total, 1 mg of RClosed-DTE-TPECM or ROpen-DTE-TPECM and 2 mg of amphiphilic lipid-PEG (DSPE-PEG$_{2000}$-maleimide) were dissolved in 1 mL of tetrahydrofuran (THF). The obtained THF solution was poured into 9 mL of deionized water under sonication with a microtip probe sonicator (XL2000, Misonix Incorporated, NY). The mixture was then sonicated for another 1 min and violently stirred in fume hood overnight at room temperature to evaporate residue THF. The NP suspension was purified by ultrafiltration (molecule weight cutoff 100,000 Da) at $3000 \times g$ for 30 min and filtered through a 0.2 μm syringe driven filter.

**Preparation of YSA-conjugated NPs.** The CYSAYPDSVPMMS peptide was used as a tumour-targeting ligand, which was modified on the RClosed NPs and ROpen NPs, respectively. Briefly, 0.2 μmol of CYSAYPDSVPMMS peptides were added into the NP suspension, followed by reaction between the thiol group of peptide and the maleimide group of PEG on the NPs under stirring for 12 h, affording RClosed-YSA NPs and ROpen-YSA NPs. The free CYSAYPDSVPMMS peptides that were not conjugated on the NPs were then removed by ultrafiltration.

**Cell culture.** Luciferase transgenic 4T1 breast cancer cells were purchased from PerkinElmer Inc. (source of parental line was from American Type Culture Collection (ATCC)). The 4T1 cancer cells were cultured in RPMI-1640 medium supplemented with 10% FBS, 10 U/mL penicillin and 10 mg/mL streptomycin. The cells were regularly checked for mycoplasma contamination and maintained in an atmosphere of 5% CO$_2$ and 95% humidified air at 37 °C.

**Animals and tumour-bearing mouse model.** All animal studies were conducted under the guidelines set by Tianjin Committee of Use and Care of Laboratory Animals, and the overall project protocols were approved by the Animal Ethics Committee of Nankai University. Male and female Sprague–Dawley rats (~300 g) as well as 6-week-old female BALB/c mice were obtained from the Laboratory Animal Center of the Academy of Military Medical Sciences (Beijing, China). To establish the xenograft 4T1 tumour-bearing mouse model, luciferase-tagged 4T1 breast cancer cells ($1 \times 10^6$) suspended in 30 μL of RPMI-1640 medium were injected subcutaneously into the right axillary space of the BALB/c mouse. After about 10 days, mice with tumour volumes of about 80−120 mm$^3$ were used subsequently.

**In vivo toxicity assessment.** Healthy BALB/c mice were randomly selected and then assigned into three groups with each group containing four mice. Two groups of mice were intravenously injected with 100 μL of RClosed-YSA NPs and ROpen-YSA NPs (1.6 mM based on DTE-TPECM molecule), respectively. The mice in the third group were untreated as a control. On day 7 post injection, all the mice in three groups were sacrificed and the blood was collected through cardiac puncture at time of sacrifice for blood chemistry analyses by Tianjin Medical University Cancer Institute and Hospital. Furthermore, on day 7, the normal organs of mice in three groups including liver and spleen were excised for histology observation. Briefly, the organs were fixed in 10% neutral buffered formalin, which were then processed routinely into paraffin, sliced at thickness of 4 μm and stained with H&E. The H&E-stained slices were imaged by a digital microscope (Leica QWin) and evaluated by three independent pathologists that were blinded to the project.

**In vivo PA imaging.** The xenograft 4T1 tumour-bearing mice were randomly selected for the following PA imaging experiment. The tumour-bearing mice were anesthetized using 2% isoflurane in oxygen, and the RClosed-YSA NPs and RClosed NPs (100 μL, 800 μM based on RClosed-DTE-TPECM) were intravenously injected into the tumour-bearing mice using a microsyringe, respectively ($n$ = 3 mice for each group). In vivo PA imaging of tumours was performed by a commercial small-animal opt-acoustic tomography system (MOST, iTheraMedical, Germany). The PA images were acquired at 700 nm before administration and at designated time intervals after injection.

**Tumour resection and intraoperative fluorescence imaging.** The xenograft 4T1 tumour-bearing mice were randomly selected for the following treatments. The tumour-bearing mice were intravenously injected with 100 μL of RClosed-YSA NPs (800 μM based on RClosed-DTE-TPECM). With the information provided by PA imaging at 4 h post injection, the tumours of mice were then resected. Briefly, tumour-bearing mice were anesthetized in an induction chamber using isoflurane. The tumour tissues were aseptically prepped and sterile instruments were employed to excise the tumours. After that, 610 nm red light (0.3 W cm$^{-2}$) was immediately irradiated at the operative incision site for up to 5 min, followed by fluorescence imaging of mice with the Maestro EX fluorescence imaging system

(CRi, Inc.) with excitation centred at 455 nm (435–480 nm) and signal collection from 500 to 720 nm in 10 nm steps. The tissues at the operative incision sites were subsequently dissected, sliced and stained with H&E. The slices obtained were examined by a digital microscope (Leica QWin) to evaluate whether there were residual tumours left behind after surgery.

**Debulking surgery and PDT of residual tumours.** The xenograft 4T1 tumour-bearing mice were randomly selected and then divided into five groups, named 'debulking surgery (DS) alone', 'DS + Light', 'DS + YSA NPs', 'DS + NPs + Light' and 'DS + YSA NPs + Light', respectively. On day 0, 100 μL of RClosed-YSA NPs (800 μM based on RClosed-DTE-TPECM) were injected into each mouse in 'DS + YSA NPs' and 'DS + YSA NPs + Light' groups via the tail vein. In addition, RClosed NPs (100 μL, 800 μM based on RClosed-DTE-TPECM) were intravenously administrated into the mice in 'DS + NPs + Light' group. At 4 h post injection, the mice in all five groups were anesthetized in an induction chamber using isoflurane. The tumour tissues were aseptically prepped, and sterile instruments were employed to debulk the tumours. The residual tumours left behind after resection were imaged by bioluminescence of the cancer cells as well as measured by a digital calliper. For bioluminescence imaging, a solution of luciferase substrate D-luciferin (150 mg kg$^{-1}$) was intraperitoneally injected into the mice, followed by imaging using the Xenogen IVIS Lumina II system. In each of five groups, the mouse bearing residual tumour with diameter of $3.5 \pm 1$ mm and thickness of $2.5 \pm 1$ mm was selected for the following experiments and finally each group contained ten mice (power > 0.8 with G*power analysis). For the mice in 'DS + Light', 'DS + YSA NPs', 'DS + NPs + Light' and 'DS + YSA NPs + Light' groups, after debulking, 610 nm red light (0.3 W cm$^{-2}$) irradiation was performed at the incision site for 5 min to convert ring-closing NPs to ring-opening NPs in the residual tumours. This was followed by white light (0.5 W cm$^{-2}$) irradiation for another 5 min on residual tumours from mice in 'DS + Light', 'DS + NPs + Light' and 'DS + YSA NPs + Light' groups to make ring-opening NPs generate ROS for PDT. The wound was then closed using surgical sutures. The tumour burden was monitored over time with bioluminescence imaging by virtue of luciferase-tagged cancer cells and the bioluminescence signals were quantitatively analysed in units of photons per second per square centimetre per steridian. The survival rates were monitored throughout the study.

**Data analysis.** The H&E-stained slices of normal organs were evaluated by three independent pathologists that were blinded to the project. For other studies, the investigators were not blinded to the group allocation. No data were excluded from the analysis. Quantitative data were expressed as mean ± standard deviation (s.d.). Statistical comparisons were made by unpaired Student's $t$-test (between two groups) and one-way ANOVA (for multiple comparisons). $P$ value < 0.05 was considered statistically significant. All statistical calculations were carried out with GraphPad Prism, including assumptions of tests used (GraphPad Software).

**Data availability.** All relevant data that support the findings of this study are available from the corresponding authors upon reasonable request.

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

## Acknowledgements

This work was supported by the NSFC (51622305, 21788102), the National Basic Research Program of China (2015CB856503), the University Grants Committee of Hong Kong (AoE/P-03/08), the Research Grants Council of Hong Kong (16301614, 16305015, N_HKUST604/14), the Innovation and Technology Commission (ITC-CNERC14SC01 and RE:ITCPD/17-9), the PCSIRT (IRT13023), the Science & Technology Project of Tianjin of China (No. 15JCYBJC29800) and the Hong Kong Scholars Program (XJ2016008). B.Z.T. is grateful for the support from the Guangdong Innovative Research Team Program of China (201101C0105067115).

## Author contributions

D.D. and B.Z.T. conceived and designed the study. J.Q. synthesized and characterized the compounds. J.Q. and C.C. performed the NP preparation and in vitro experiments. C.C., X. Z. and S.J. performed the in vivo experiments. X.H. provided technical assistance with PA imaging. J.Q., C.C., R.T.K.K., J.W.Y.L., D.D. and B.Z.T. analysed the data and participated in the discussion. J.Q., C.C., D.D. and B.Z.T. contributed to the writing of this paper.

## Additional information

**Competing interests:** The authors declare no competing interests.

