## [Peer Review File · Nature Communications]

Reviewers' Comments:

Reviewer #1:

Remarks to the Author:

In this study, Ji Qi et al. developed a light-driven function-transformable nanoparticles for pre-operative photoacoustic imaging and intraoperative fluorescence imaging and photodynamic therapy. It is a very interesting study to set a switch for converting the function of the nanoparticle. The chemistry is very solid. The ring-nanoparticle showed excellent PA property when being excited at 680 nm. The conversion from close to open form is very efficient. The open form NP can be excited at 435 nm with emission at 550 nm. Then the authors applied the NP in a tumor xenografted mouse to model using 4T1 cells to further demonstrate the in vivo application of the NP.

1. Starting from a very smart design, the manuscript basically presented a complete story and potentially useful strategy.
2. Along with the success on chemistry, the biological part is exaggerated. Some contents are misleading, and the shortcomings of the NP are not fully presented.
3. The whole-body pharmacokinetics is not there for either an imaging probe or therapeutic agent after intravenous injection, which is very important to make sound judgement of the biological application of the nanomaterials.
4. The author mentioned PA imaging is with deep tissue penetration and high spatial resolution in several places. However, the tissue penetration of PA imaging is very limited compared with other popular clinical used imaging methods including MRI, CT, PET, even ultrasound. Moreover, the high spatial resolution of PA is only limited to microscopic PA imaging. With the set-up used in this manuscript, neither deep tissue penetration nor high spatial resolution was identified. In fact, without knowing the tumor position in advance, it is very challenging for PA imaging to localize the tumor region.
5. The excitation and emission of the NP when being used as a fluorescence dye is not ideal for in vivo application. NIR dyes are still the main stream for in vivo fluorescence optical imaging.
6. EPR effect may be adequate for tumor delivery of nanoformulations. However, it is not reliable for tumor detection since the targeting is not specific. Preclinical xenograft is so different from the real malignant tissue in patients. This may be partially the reason why quite a lot of nanoparticles showed excellent efficacy in animal experiments but no in clinical trials.

Reviewer #2:

Remarks to the Author:

Qi and coworkers describe a novel contrast agent with utility in both fluorescence and photoacoustic imaging. The reason this is important is that most other approaches to multimodal imaging take a contrast agent and tag it with a second contrast agent. This generally results in a product that has only marginal performance in either modality. Because this approach is switchable—and can be switched many times with no loss of signal—it can offer excellent performance in BOTH modalities. This is especially important for surgery where the deep penetration of photoacoustics is useful for pre-surgical staging. This can then be coupled with fluorescence for superficial imaging of tumor debulking. This is an important paper for the field because it shows a novel approach to activatable photoacoustic contrast agents. While the field has seen a variety of chemically-sensitive probes, these optically-switchable tools are less common—especially with good performance in two modalities and with a therapeutic component as shown here. For these reasons, I am enthusiastic about publication in the journal.

Nevertheless, the paper has some limitations that need to be addressed. I have detailed those more carefully below. After these changes have been incorporated, I recommend publication in Nature Communications.

1. The text should be proofread by a native English speaker.

2. The MS and NMR spectra are very clean and convincing.
3. Fig. 1c (or 1b): Why not plot photoacoustic signal on the other y axis to show correlation between PA and fluorescence?
4. The authors suggest that the mechanism is FRET. Thus, one wonders why there is no change in the peak wavelength of emission. Rather, there seems to be a change in the intensity of fluorescence, but FRET is well known to induce a change in the emission maximum.
5. Can the authors comment on the Forrester distance for these FRET donor/acceptor pair?
6. Fig. 3f: The authors show that the absorption is recoverable across multiple cycles. Does the fluorescence and photoacoustic signal show similar behavior.
7. Fig. 4H is confusing. Consider replacing with a table.
8. Missing citations to photoacoustic imaging:
 - a. <http://pubs.rsc.org/en/content/articlehtml/2015/an/c5an00207a>
 - b. <http://onlinelibrary.wiley.com/doi/10.1002/wnan.1404/full>
9. Figure 4 is confusing because I and I0 have different meaning in different panels. In 4e, it is presence of absence of ROS. This shows that only ICG is sensitive to ROS, which doesn't make sense. I and I0 are not defined in 4i. In 4j, it is before/after irradiation.
10. Page 13, "MB is also a star molecular for PA imaging." Not really. It has only average brightness, but it is commonly used.
11. Fig. 4e: SPNs have also been shown to be sensitive to RONS:
 - a. <https://www.ncbi.nlm.nih.gov/pmc/articles/PMC3947658/>
12. Fig. 4i. Although not explicitly necessary, it would be impressive to also include a control with DHEA scavenger to further prove ROS generation.
13. Fig. 4i. Although not explicitly necessary, it would be impressive to measure increased ROS in the plasma of animals "DS + NPs + Light" in the group relative to DS + NP. Showing an independent measurement would help your case.
14. Page 15. Please state the concentration of NPs injected i.v. here.
15. Fig. 5h: Define DS in caption.
16. Figure 6 is very impressive.
17. Can this photoswitchable behavior still occur at greater depths? How much power do you need to create the shift? Showing switching through a 1 cm piece of agar or chicken breast would be impressive.

Reviewer #3:

Remarks to the Author:

The manuscript by Qi et al. reported a new kind of light-driven transformable agent for photoacoustic, fluorescence imaging and photodynamic therapy by turning the energy balance of photophysics under external light stimuli. The molecular structure, photophysical properties and photoreversibility of the organic molecules were studied. In vitro fluorescence, photoacoustic and ROS generation were investigated, which matched well with the design principle. The closed-ring molecule with good conjugation showed NIR absorption and no radiative decay, thus it could be used for in vivo tumor detection. After surgical resection of the tumors, the molecule changed to open-ring form under red light irradiation, and became a sensitive aggregation-induced emission fluorescent agent to inspect the surgery outcomes. For the residual tumors after debulking surgery, the fluorescent agent could also be used for photodynamic therapy (PDT) to improve the survival rate. The investigation is interesting and important for biomedical imaging/therapy community. The manuscript is well organized and the overall quality is good. Therefore, this manuscript can be published in Nature Communications after addressing the following issues.

Questions:

1. The authors have compared the fluorescence and photoacoustic properties of the prepared organic agent with a semiconducting polymer and organic dye (MB). The authors should provide absorption spectrum of the semiconducting polymer and MB in the supporting information, as they may also have some influence on the photoacoustic performance.

2. It seems that the photoreversibility property of the nanoparticles is slower than the solution. Could the authors give some explanation about this?
3. The ¹³C NMR of compound DTE-2Br (2) is a little strange, so I advise the authors to check and measure it again. The authors should add the basic characterization (H-NMR and HRMS) of RClosed-DTE-TPECM in the supporting information.
4. The authors said "In the ring-closing form, Förster resonance energy transfer (FRET) from TPECM to closed-ring DTE". Could the authors show some spectral data to explain the FRET phenomenon?
5. The authors claimed that "there is no detectable photoluminescence (PL) emission of the ring-closing isomer in both THF solution and aggregation form". I suggest the authors to measure the PL spectra of the ring-closed compound and NPs in the NIR region (up to 1000 nm).
6. The authors should provide the absorption spectrum of the open-form molecule in the supporting information.
7. The authors use red light (610 nm) to change RClosed NPs to ROpen NPs, so would the red-light irradiation cause any thermal effect? I suggest the authors to measure the photothermal effect by using the red-light irradiation as mentioned in the manuscript. In addition, the authors should add another MTT data (RClosed NPs+610 nm light) in Supplementary Figure 23.
8. In the manuscript, the authors used white light for PDT study, so what's the emission spectrum of the light source?
9. How about the in vivo biocompatibility of the nanoparticles? Could the authors perform some measurements, such as serum biochemistry assay and complete blood count of the mice?
10. Citation for DFT calculation should be added in the Reference part. I advise the authors to add some references related to fluorescence and/or PA imaging guided PDT, e.g., Nat. Commun., 2014, 5, 4596; Adv. Sci., 2017, 4 (10), 1700085.

Responses to the Comments and Suggestions of Reviewer #1

Comments:

In this study, Ji Qi et al. developed a light-driven function-transformable nanoparticles for pre-operative photoacoustic imaging and intraoperative fluorescence imaging and photodynamic therapy. It is a very interesting study to set a switch for converting the function of the nanoparticle. The chemistry is very solid. The ring-nanoparticle showed excellent PA property when being excited at 680 nm. The conversion from close to open form is very efficient. The open form NP can be excited at 435 nm with emission at 550 nm. Then the authors applied the NP in a tumor xenografted mouse to model using 4T1 cells to further demonstrate the in vivo application of the NP.

Q1. Starting from a very smart design, the manuscript basically presented a complete story and potentially useful strategy.

Response: We sincerely thank the reviewer for your careful reading and positive feedback to our work.

Q2. Along with the success on chemistry, the biological part is exaggerated. Some contents are misleading, and the shortcomings of the NP are not fully presented.

Response: After carefully reading the reviewer's comments and checking our manuscript, we strongly agree with the reviewer that the biological part is exaggerated. Therefore, according to the reviewer's suggestion, we have significantly changed/deleted the contents that are misleading, and fully presented the shortcomings of our NPs in the revised manuscript. Please refer to the responses to Q4 and Q5. Besides, we have carefully checked the whole manuscript, and several other overstatements in the biological part have also been deleted and changed (highlighted in red for your reference) in the revised manuscript.

Q3. The whole-body pharmacokinetics is not there for either an imaging probe or therapeutic agent after intravenous injection, which is very important to make sound judgement of the biological application of the nanomaterials.

Response: We thank the reviewer very much for this important constructive suggestion. According to the reviewer's suggestion, we have performed the *in vivo* pharmacokinetics study and the corresponding data and results have been added in the revised manuscript. It is worthy to note that according to the reviewer's comment in Q6, we have modified our function-transformable NPs with a tumour targeting ligand (a peptide with the sequence of YSAYPDSVPMMS), affording YSA-conjugated ring-closing NPs (RClosed-YSA NPs) and YSA-conjugated ring-opening NPs (ROpen-YSA NPs). The YSAYPDSVPMMS (named YSA in short) peptide was selected as the targeting ligand as it can selectively and tightly bind to EphA2 protein, which is a transmembrane receptor tyrosine kinase overexpressed in many cancer cells as well as tumour blood vessels including 4T1 mammary adenocarcinoma (*Chem. Sci.* **8**, 2191–2198 (2017); *Clin. Cancer Res.* **19**, 128–137 (2013); *Oncogene* **21**, 7011–7026 (2002)). We have demonstrated that the YSA modification does not influence any of the NP properties in terms of PA and fluorescence properties, ROS generation capacity, and reversible photochromism. Furthermore, we re-performed nearly all of the cellular and *in vivo* experiments to assess the feasibility and advantage of YSA-conjugated NPs in improving cancer surgery outcomes. The new pre-operative photoacoustic (PA) imaging result reveals that RClosed-YSA NPs can be accumulated in tumour tissues more effectively than RClosed NPs without YSA conjugation (for example, ~1.8 times higher at 4 h post-injection). Additionally, the PDT efficacy of ROpen-YSA NPs transformed from RClosed-YSA NPs by red light irradiation within residual tumours post surgery is better than that of the converted ROpen NPs. In a word, the new cellular and *in vivo* data verify that YSA-conjugated NPs can serve as a more efficient imaging probe/therapeutic agent for the application in cancer surgery.

As a consequence, YSA-conjugated NPs is the staple imaging probe/therapeutic agent in our revised manuscript, and it is important to investigate their *in vivo* pharmacokinetics. Although the ROpen-YSA NPs have bright yellow emission, it has been reported that fluorescence strategy is not ideal to quantitatively analyze the pharmacokinetics and biodistribution of the probes *in vivo* (*Adv. Funct. Mater.* **16**, 1299–1305 (2006)). Since it is well accepted that radiolabelling is a routine and reliable method to track injected species in pharmacokinetic and biodistribution studies (*Nat. Commun.* **4**, 2016 (2013); *Nat. Nanotechnol.* **2**, 47–52 (2007); *Adv. Funct. Mater.* **16**, 1299–1305 (2006); *J. Control. Release* **235**, 306–318

(2016)), we selected radiolabelling strategy and used a radioactive nuclide, iodine-125 (^{125}I), to label RClosed-YSA NPs through the reaction between ^{125}I and the tyrosine (Y) residues of YSA peptide on the NPs. The radiochemical purity of ^{125}I -labelled RClosed-YSA NPs is higher than 99%, which does not change upon keeping the NPs in saline for 3 days, revealing the high radiolabelling stability.

The *in vivo* pharmacokinetics of ^{125}I -labelled RClosed-YSA NPs was studied using healthy Sprague–Dawley rats ($n = 6$ rats). After ^{125}I -labelled RClosed-YSA NPs were intravenously injected into the rats, the blood samples were collected at designated time intervals (0.083, 0.25, 0.5, 1, 1.5, 2, 3, 4, 6, 8, 10, 12, and 24 h) and counted for ^{125}I radioactivity with a gamma counter. The result has been added as new Fig. 5f in the revised manuscript, which reveals the blood circulation behavior of ^{125}I -labelled RClosed-YSA NPs. The pharmacokinetic parameters including circulation half-life ($t_{1/2}$), $\text{AUC}_{(0-\infty)}$, CL, Vd, C_{max} and MRT of the RClosed-YSA NPs are determined to be 6.21 ± 0.39 h, $205.31 \pm 19.94 \mu\text{g h mL}^{-1}$, $24.47 \pm 2.38 \text{ mL kg}^{-1} \text{ h}^{-1}$, $143.03 \pm 12.12 \text{ mL kg}^{-1}$, $49.97 \pm 3.91 \mu\text{g mL}^{-1}$, and 5.85 ± 0.07 h, respectively. The complete pharmacokinetic data have also been added in Supplementary Table 2 in the revised Supplementary Information.

Besides, the biodistribution of RClosed-YSA NPs in xenograft 4T1 tumour-bearing mice was investigated as well by virtue of ^{125}I labelling. After ^{125}I -labelled RClosed-YSA NPs were administrated into 4T1 tumour-bearing mice through the tail vein, the time-dependent biodistributions of the NPs in blood, tumour and various major organs of mice were quantitatively analyzed by gamma scintillation counting ($n = 6$ mice for each time point). The *in vivo* biodistribution result has been added as new Fig. 5g in the revised manuscript, which indicates that the RClosed-YSA NPs rapidly leave the bloodstream and enter most organs, and the levels of the NPs in all of the tissues significantly decrease after 8 hours. Due to the reticuloendothelial system (RES) and mononuclear phagocyte system (MPS) uptake (*Mol. Pharmaceutics* **5**, 496–504 (2008); *Nat. Nanotechnol.* **12**, 1023–1025 (2017)), high accumulations of the NPs in liver, spleen and bone marrow are found. Importantly, thanks to both the "active" (YSA-EphA2 interaction) and "passive" (the enhanced permeability and retention (EPR) effect of nanomaterials) tumour targeting capabilities, the RClosed-YSA NPs can be largely enriched in tumour tissue with maximum tumour uptake of $\sim 7.3\% \text{ ID g}^{-1}$ occurring at 4 h post-injection.

In the revised manuscript, we have added a new part "*In vivo* pharmacokinetics and biodistribution." in Results section (line 4-23, page 18 and line 1-11, page 19). Moreover, the corresponding experimental procedures have also been added in the Methods section in the revised Supplementary Information.

Q4. The author mentioned PA imaging is with deep tissue penetration and high spatial resolution in several places. However, the tissue penetration of PA imaging is very limited compared with other popular clinical used imaging methods including MRI, CT, PET, even ultrasound. Moreover, the high spatial resolution of PA is only limited to microscopic PA imaging. With the set-up used in this manuscript, neither deep tissue penetration nor high spatial resolution was identified. In fact, without knowing the tumor position in advance, it is very challenging for PA imaging to localize the tumor region.

Response: We agree with the reviewer that the statements regarding to the deep tissue penetration and high spatial resolution of PA imaging in our previous manuscript are exaggerated, and even incorrect. PA imaging is a hybrid imaging modality that combines optical excitation with ultrasonic detection, thus permitting imaging beyond the optical diffusion limit. As compared to traditional optical imaging techniques, PA imaging can provide relatively deeper tissue penetration with centimeter-scale imaging depth (*Nat. Biomed. Eng.* **1**, 0071 (2017); *Nat. Nanotechnol.* **9**, 233–239 (2014); *Nat. Biotechnol.* **24**, 848–851 (2006)). However, as the reviewer said, the tissue penetration of PA imaging is still very limited compared with other popular clinical used imaging methods including MRI, CT, PET, and even ultrasound.

Each imaging modality has its own advantages and disadvantages. Over recent years, there have been many technological advances in PA imaging instrumentation, data processing, contrast agents, as well as biomedical applications, which imply that PA imaging is a promising technique that has its intrinsic strengths (*Nat. Med.* **18**, 1297–1302 (2012); *Nat. Methods* **13**, 639–650 (2016); *Trends Biotechnol.* **29**, 213–221 (2011)). It is expected that with the joint efforts from physicists, optical engineers, chemists, biomedical engineers and clinicians, PA imaging would find its applications in the clinic. As the chemists or materials scientists, we aim to develop smart/advanced PA imaging agents that meet the requirements of the basic working principle of PA imaging. However, we are not good at developing technology and instrument to

improve the tissue penetration depth and spatial resolution of PA imaging. Thereby, in our manuscript, a commercial small-animal opt-acoustic tomography system (MOST, iTheraMedical, Germany) was employed, just as a proof-of-concept to verify that our ring-closing NPs can give good performance in *in vivo* PA imaging of tumours.

We apologize for the overstatements of PA imaging in our previous manuscript. In the revised version, we have deleted and changed all the exaggerated statements regarding to the deep tissue penetration and high spatial resolution of PA imaging. The main changes are summarized in the following Table R1.

Table R1. Main changes regarding to PA imaging

	In the previous manuscript	In the revised manuscript	
Introduction	Accordingly, the integration of fluorescence and PA imaging modes decidedly enables precise diagnostic outcome by virtue of both superb sensitivity and spatial resolution.	Accordingly, the integration of fluorescence and PA imaging modes decidedly enables precise diagnostic outcome by virtue of high sensitivity and imaging depth beyond the optical diffusion limit.	Line 2-4, page 3
Results	As the key advantage of PA imaging lies in the deep tissue penetration and high spatial resolution, it is potent to offer in vivo 3D information on the location and size of tumours before surgery. To study the utility of our NPs in in vivo PA imaging of tumours, xenograft 4T1 tumour-bearing mice were employed.	As PA technique permits imaging that surpasses the limit of optical diffusion, compared with fluorescence imaging, PA imaging could offer relatively deeper information on the tumours in vivo before surgery. As a proof-of-concept, a commercial small-animal opt-acoustic tomography system (MOST) was used to study the utility of RClosed-	Line 11-15, page 21

		YSA NPs in in vivo PA imaging of tumours.	
Discussion	Preoperative imaging requires displaying the size and location of a 3D tumour with good spatial resolution. PA imaging technique is hence very promising for preoperative diagnosis, as it is advantageous to offer 3D information of deep tumours with anatomical resolution.	PA imaging holds the potential for preoperative diagnosis, as it can offer deeper tumour information as compared to conventional optical imaging techniques.	Line 11-12, page 27

Q5. The excitation and emission of the NP when being used as a fluorescence dye is not ideal for in vivo application. NIR dyes are still the main stream for in vivo fluorescence optical imaging.

Response: We thank the reviewer for pointing it out. And we agree with the reviewer very much that near-infrared (NIR) dyes are the main stream for *in vivo* fluorescence imaging and the excitation and emission of our fluorescent ring-opening NPs are not ideal for *in vivo* application. As shown in Fig. 6e in the revised manuscript, our function-transformable NPs show good performance in detecting the residual tumours left behind after tumour resection surgery with their light-up yellow fluorescence (high signal-to-background ratio of ~7.1). This is because the yellow fluorescent NPs are applied for intraoperative fluorescence imaging during surgery, but not used for noninvasive *in vivo* fluorescence imaging, which to a certain extent address the low tissue penetration issue of visible light. It has also been reported that the visible light emitters (*e.g.*, green-yellow emitters) can favor the intraoperative fluorescence diagnosis (*Nat. Med.* **17**, 1315–1319 (2011); *Nat. Rev. Cancer* **13**, 653–662 (2013)). Nevertheless, NIR dyes are undoubtedly a better alternative to visible emitters even in the *in vivo* application of intraoperative fluorescence imaging.

On the other hand, in our manuscript, we aim to put forward a new concept of function-transformable optical agent with maximized effectiveness of each function. As a proof-of-

concept, the ring-opening NPs are firstly designed to be yellow emissive in order to avoid complicated synthetic steps. In the future work, we will focus on developing function-transformable optical agents with fluorescence imaging in the NIR spectral region to better meet the requirements of practical *in vivo* applications.

In the revised manuscript, we have presented the shortcoming of our system and added "However, the current work still has limitation, as the excitation and emission of the ring-opening NPs are not ideal for practical *in vivo* applications. Future work will focus on developing function-transformable optical agents with fluorescence imaging in the NIR spectral region." in the Discussion section at line 19-22 page 28.

Q6. EPR effect may be adequate for tumor delivery of nanoformulations. However, it is not reliable for tumor detection since the targeting is not specific. Preclinical xenograft is so different from the real malignant tissue in patients. This may be partially the reason why quite a lot of nanoparticles showed excellent efficacy in animal experiments but no in clinical trials.

Response: We agree with the reviewer that EPR effect may be adequate for tumor delivery of nanoformulations; however, it is not reliable for tumor detection since the targeting is "passive" and not specific. It is also true that the unreliable EPR effect may be partially the reason why quite a lot of nanoparticles showed excellent efficacy in animal experiments but no in clinical trials. Therefore, in the revised manuscript, we have demonstrated that our NPs are able to be facilely surface functionalized with tumour targeting ligands, realizing more effective tumour targeting and uptake as compared to EPR effect alone.

As a proof-of-concept, we have modified our function-transformable NPs with a tumour targeting ligand (a peptide with the sequence of YSAYPDSVPMMS), affording YSA-conjugated ring-closing NPs (RClosed-YSA NPs) and YSA-conjugated ring-opening NPs (ROpen-YSA NPs). The YSAYPDSVPMMS (named YSA in short) peptide was selected as the targeting ligand as it can selectively and tightly bind to EphA2 protein, which is a transmembrane receptor tyrosine kinase overexpressed in many cancer cells as well as tumour blood vessels including 4T1 mammary adenocarcinoma (*Chem. Sci.* **8**, 2191–2198 (2017); *Clin. Cancer Res.* **19**, 128–137 (2013); *Oncogene* **21**, 7011–7026 (2002)). In our revised manuscript, we have demonstrated that the 4T1 cancer cells used in our study overexpress EphA2 proteins *via* western blotting (new

Fig. 5c in the revised manuscript). To prepare YSA-conjugated NPs, we firstly synthesized CYSAYPDSVPMMS peptide (new Fig. 5a in the revised manuscript) with a terminal thiol group in cysteine (C) *via* standard 9-fluorenylmethoxycarbonyl (Fmoc) solid-phase peptide synthesis (SPPS), which was characterized by LC-MS and HRMS (Supplementary Figs 27 and 28 in the revised Supplementary Information). The RClosed NPs and ROpen NPs were then modified with CYSAYPDSVPMMS peptide through the coupling reaction between the thiol group of peptide and the maleimide group of PEG on the NPs (1,2-distearoyl-*sn*-glycero-3-phosphoethanolamine-*N*-[(polyethylene glycol)-2000]-maleimide (DSPE-PEG₂₀₀₀-maleimide) was used as the encapsulation matrix of NPs), yielding RClosed-YSA NPs and ROpen-YSA NPs, respectively. It is calculated that there are ~3700 YSA peptides on average conjugated on each NP (the detailed calculation has been added in the revised Supplementary Information). The mean size of YSA-conjugated NPs is ~68 nm determined by DLS, which is similar to that of the NPs without YSA (~65 nm). It is also verified that the YSA modification does not influence any of the NP properties in terms of PA and fluorescence properties, ROS generation capacity, and reversible photochromism. The next *in vitro* cellular experiments (new Fig. 5e and Supplementary Figs 29-32 in the revised Supplementary Information) not only manifest the remarkable target effect by YSA modification on NP surface, but also verify that the RClosed-YSA NPs can be facilely transformed to ROpen-YSA NPs in biological environment, triggered by red light irradiation.

The RClosed-YSA NPs were then used as the staple imaging probe/therapeutic agent in the entire study regarding to cancer surgery, including pre-operative PA imaging as well as intraoperative fluorescence imaging and PDT of residual tumours after debulking surgery. These results have been added as new Figs 6 and 7 in the revised manuscript, which reveal that RClosed-YSA NPs give better performance in improving cancer surgery outcomes compared with those without YSA modification, highlighting the importance of introducing active targeting ligand on the NPs.

In the revised manuscript, we have added a new part "**NP surface modification with a targeting ligand.**" in Results section (line 8-22, page 16, line 1-23, page 17, and line 1-3, page 18). In addition, the content in "**Improvement of cancer surgery outcomes.**" part in Results section has also been significantly changed due to the involvement of YSA-conjugated NPs, which has been highlighted in red in the revised manuscript for your reference. Besides, the

corresponding experimental procedures have also been added in the Methods section in the revised Supplementary Information.

Responses to the Comments and Suggestions of Reviewer #2

Comments:

Qi and coworkers describe a novel contrast agent with utility in both fluorescence and photoacoustic imaging. The reason this is important is that most other approaches to multimodal imaging take a contrast agent and tag it with a second contrast agent. This generally results in a product that has only marginal performance in either modality. Because this approach is switchable—and can be switched many times with no loss of signal—it can offer excellent performance in BOTH modalities. This is especially important for surgery where the deep penetration of photoacoustics is useful for pre-surgical staging. This can then be coupled with fluorescence for superficial imaging of tumor debulking. This is an important paper for the field because it shows a novel approach to activatable photoacoustic contrast agents. While the field has seen a variety of chemically-sensitive probes, these optically-switchable tools are less common—especially with good performance in two modalities and with a therapeutic component as shown here. For these reasons, I am enthusiastic about publication in the journal.

Nevertheless, the paper has some limitations that need to be addressed. I have detailed those more carefully below. After these changes have been incorporated, I recommend publication in Nature Communications.

Q1. The text should be proofread by a native English speaker.

Response: We sincerely thank the reviewer for your careful reading and positive feedback to our work. According to the reviewer's suggestion, we have invited a native English speaker to proofread our manuscript carefully. The corresponding changes have been made in the revised manuscript.

Q2. The MS and NMR spectra are very clean and convincing.

Response: We thank the reviewer for this positive comment.

Q3. Fig. 1c (or 1b): Why not plot photoacoustic signal on the other y axis to show correlation between PA and fluorescence?

Response: In our manuscript, Figs 1b shows the photoluminescence (PL) spectra of ROpen-DTE-TPECM (ring-opening molecule) in THF/water mixture with various water fractions, and Fig. 1c displays the Plot of I/I_0 versus water fraction. I_0 and I are the peak PL intensities of ROpen-DTE-TPECM (10 μ M) in pure THF and THF/water mixtures, respectively. As a consequence, the result in Figs 1b and 1c demonstrates the aggregation-induced emission (AIE) feature of the fluorescent ring-opening molecule. As the ring-opening molecule has no PA signal, we cannot plot PA signal on the other y axis.

Q4. The authors suggest that the mechanism is FRET. Thus, one wonders why there is no change in the peak wavelength of emission. Rather, there seems to be a change in the intensity of fluorescence, but FRET is well known to induce a change in the emission maximum.

Response: According to the mechanism of FRET, when the energy transfers from the donor emitter to the acceptor emitter, there should be changes in the emission spectra of both the donor and acceptor. In our manuscript, for the ring-closing molecule (RClosed-DTE-TPECM), we have suggested that FRET occurs from the fluorescent TPECM donors to the non-emissive ring-closing DTE core (acceptor). This is because several literatures have reported that the ring-closing DTE often acts as a non-emissive acceptor to quench the donor emission by FRET process (*Chem. Rev.* **114**, 12174–12277 (2014); *Adv. Mater.* **25**, 378–399 (2013)). According to the reviewer's comment, we have carefully measured the emission property of RClosed-DTE-TPECM. It is verified that the yellow donor (TPECM) fluorescence is almost vanished, meeting the mechanism of FRET. Furthermore, there is no emission of RClosed-DTE-TPECM detected in either the solution state or nanoparticle form even extending the wavelength to 1200 nm (see Supplementary Fig 19 in the revised Supplementary Information), indicating negligible increase in acceptor (ring-closing DTE) emission. In a word, the FRET mechanism regarding to ring-closing DTE is suggested according to the literatures; however, the ring-closing DTE itself is

non-emissive, thus leading to no elevation of acceptor emission. In the revised manuscript, we have cited "*Chem. Rev.* **114**, 12174–12277 (2014)" and "*Adv. Mater.* **25**, 378–399 (2013)" as refs. 25 and 26 to support the claim "there is no detectable photoluminescence (PL) emission of the ring-closing isomer in both THF solution and aggregation form even extending the wavelength to 1200 nm (Supplementary Fig. 19), because of the intramolecular energy transfer from the fluorescent TPECM donors to the non-emissive ring-closing DTE core (acceptor)".

Q5. Can the authors comment on the Förster distance for these FRET donor/acceptor pair?

Response: According to the equation: $R_0 = 0.211 (n^{-4} Q_d k^2 J)^{1/6} \text{Å}$, where n is the refractive index of the medium, Q_d is the fluorescence quantum yield of the donor, J is the overlap integral between the donor emission spectrum and the acceptor absorbance spectrum, and k^2 is the dipole orientation factor (assumed to be 2/3 based on random orientation distribution of donor and acceptor molecules). The Förster distance (R_0) is thus calculated to be 3.4 nm, which is similar to the data reported in literatures (*Chem. Commun.* **51**, 4036–4039 (2015); *Adv. Mater.* **22**, 1602–1606 (2010)).

Q6. Fig. 3f: The authors show that the absorption is recoverable across multiple cycles. Does the fluorescence and photoacoustic signal show similar behavior.

Response: Fig. 3f shows the reversible absorption property of the NPs during ten circles of visible (610 nm, 0.3 W cm⁻²)/UV light (365 nm, 0.1 W cm⁻²) irradiation processes. According to the reviewer's suggestion, we have also measured the reversible fluorescence and PA properties of our NPs during ten circles of visible (610 nm, 0.3 W cm⁻²)/UV light (365 nm, 0.1 W cm⁻²) irradiation processes. The results have been added as Supplementary Fig. 21 in the revised manuscript, which indicate that the ring-closing and ring-opening NPs can convert reversibly by alternating UV/visible light irradiation with negligible interference on the emission and PA properties during 10 circles. In the revised manuscript, the corresponding change has been made at line 15-18, page 10.

Q7. Fig. 4H is confusing. Consider replacing with a table.

Response: According to the reviewer's suggestion, in the revised manuscript, we have deleted Fig. 4H. Instead, the optical and PA properties of various agents were listed in a table, which has been added as new Supplementary Table 1 in the revised version.

Q8. Missing citations to photoacoustic imaging:

a. <http://pubs.rsc.org/en/content/articlehtml/2015/an/c5an00207a>

b. <http://onlinelibrary.wiley.com/doi/10.1002/wnan.1404/full>

Response: We thank the reviewer for this valuable comment. The missing citations to PA imaging are indeed important. In the revised manuscript, "*Analyst* **140**, 3731-3737 (2015)" and "*WIREs Nanomed. Nanobiotechnol.* **9**, e1404 (2017)" have been cited as refs 17 and 8.

Q9. Figure 4 is confusing because I and I₀ have different meaning in different panels. In 4e, it is presence of absence of ROS. This shows that only ICG is sensitive to ROS, which doesn't make sense. I and I₀ are not defined in 4i. In 4j, it is before/after irradiation.

Response: The reviewer is correct in pointing out that Figure 4 is confusing as *I* and *I*₀ have different meaning in different panels. Therefore, in the revised manuscript, we have used *A* and *A*₀ to replace *I* and *I*₀, respectively, in revised Figure 4e, where *A* and *A*₀ are the absorption intensity at 680 nm of RClosed NPs, SPNs, MB and ICG in the presence and absence of RONS (400 μM), respectively. In addition, in the revised version, we have also added "*I*₀ and *I* are the PL intensity of DCF at 525 nm before and after light irradiation at designated time intervals in both (i) and (j)." in the caption of Figure 4.

Q10. Page 13, "MB is also a star molecular for PA imaging." Not really. It has only average brightness, but it is commonly used.

Response: We thank the reviewer for pointing it out. In the revised manuscript, we have changed "MB is also a star molecule for PA imaging" to "MB is also a commonly used molecule for PA imaging" at line 6, page 13.

Q11. Fig. 4e: SPNs have also been shown to be sensitive to RONS:

a. <https://www.ncbi.nlm.nih.gov/pmc/articles/PMC3947658/>

Response: Yes, in the paper by Pu *et al.* (Semiconducting polymer nanoparticles as photoacoustic molecular imaging probes in living mice. *Nat. Nanotechnol.* **9**, 233–239 (2014)), they reported a kind of RONS imaging PA probe. In this paper, they linked a RONS-sensitive cyanine dye derivative (IR775S) to achieve ratiometric PA imaging. The SPN itself had high stability towards RONS. After linking IR775S to the surface of the SPN, the IR775S-linked SPNs exhibited dual PA peaks. One (700 nm) was derived from the semiconducting polymer core, and the other one (820 nm) was from the cyanine dye surface. When exposure to certain RONS, the surface IR775S dye would be destroyed, but the core semiconducting polymer had no changes. As a result, the PA signal (820 nm) from IR775S disappeared, while the PA signal (700 nm) from semiconducting polymer remained. They applied this nanoplatform for *in vitro* and *in vivo* PA imaging of RONS. This paper has been cited as ref. 31 in the revised manuscript.

Q12. Fig. 4i. Although not explicitly necessary, it would be impressive to also include a control with DHEA scavenger to further prove ROS generation.

Response: In the revised manuscript, we employed sodium azide (NaN_3), which is known to selectively scavenge singlet oxygen ($^1\text{O}_2$) (*Adv. Mater.* **29**, 1606167 (2017); *Photochem. Photobiol.* **85**, 1177–1181 (2009)), for the purpose that the reviewer suggested. Sodium azide (NaN_3) was selected as it can not only prove the ROS generation, but more importantly, demonstrate whether the produced ROS is $^1\text{O}_2$ or not. In this experiment, the 4T1 cancer cells were incubated with RClosed-YSA NPs (8 μM based on RClosed-DTE-TPECM) for 4 h, followed by 610 nm red light (0.3 W cm^{-2}) irradiation for 5 min in order to convert RClosed-YSA NPs to ROpen-YSA NPs. Alternatively, the NP-treated cells were co-treated with NaN_3 (10 mM). ROS indicator 2',7'-dichlorodihydrofluorescein diacetate (DCF-DA) (10 μM) was then loaded into the cells. After 5 min incubation, the cells were washed with $1 \times \text{PBS}$ and exposed to white light (0.25 W cm^{-2}) for 2 min. Afterward, the cells were washed with $1 \times \text{PBS}$ and

immediately imaged by CLSM. For DCF detection, excitation at 488 nm and signal collection at 520 ± 10 nm were adopted.

The result has been added as new Supplementary Fig. 32 in the revised Supplementary Information. As shown in Supplementary Fig. 32a, there is very low fluorescence of DCF inside the transformed ROpen-YSA NP-loaded cells without white light exposure. Nevertheless, upon irradiation by white light for 2 min, strong green fluorescence can be clearly observed within the transformed ROpen-YSA NP-loaded cells (Supplementary Fig. 32b), suggesting effective intracellular ROS production by “transformed ROpen-YSA NPs + white light”. After the cells were co-treated with NaN_3 , a singlet oxygen scavenger, the DCF fluorescence in the “transformed ROpen-YSA NPs + white light”-treated cells is significantly reduced (Supplementary Fig. 32c), revealing that the ROS generated by “transformed ROpen-YSA NPs + white light” is mainly singlet oxygen.

Q13. Fig. 4i. Although not explicitly necessary, it would be impressive to measure increased ROS in the plasma of animals “DS + NPs + Light” in the group relative to DS +NP. Showing an independent measurement would help your case.

Response: According to the reviewer's suggestion, we have tried to measure the ROS levels in the plasma of the mice in "debulking surgery (DS) + NPs + Light" and "DS + NPs" group. However, no ROS elevation in the plasma of mice in "DS + NPs + Light" group was detected. This would be because for the treatment of "DS + NPs + Light", ROS is generated *in situ* within the residue tumour by "NPs + Light". Although some ROS molecules may penetrate into the blood *via* blood vessels in the tumour, they are short-lived species and would be hard to be detected in the plasma.

Q14. Page 15. Please state the concentration of NPs injected i.v. here.

Response: According to the reviewer's suggestion, the concentration of the NPs (100 μL , 800 μM based on RClosed-DTE-TPECM) has been added in the revised manuscript, which is now at line 18-19, page 21.

Q15. Fig. 5h: Define DS in caption.

Response: According to the reviewer's suggestion, in the revised manuscript, we have defined DS as “debulking surgery” in the caption of Fig. 7a, which is Fig. 5h in the previous version.

Q16. Figure 6 is very impressive.

Response: We thank the reviewer for this positive comment.

Q17. Can this photoswitchable behavior still occur at greater depths? How much power do you need to create the shift? Showing switching through a 1 cm piece of agar or chicken breast would be impressive.

Response: We thank the reviewer for this constructive suggestion. According to the reviewer's suggestion, we have monitored the change in absorption spectrum of the RClosed NPs under red light through different thicknesses of chicken breast to evaluate the switching property in large depths. We have measured the absorption spectra of RClosed NPs before and after red light (610 nm, 0.3 W cm^{-2}) irradiation through different thicknesses of chicken breast (0, 1, 2, 3, 5 and 10 mm) for 0.5 h. The result has been added as new Supplementary Fig. 22 in the revised Supplementary Information, which reveals that the RClosed NPs show good switching property. Even if the red light irradiation is through a 1 cm thickness of chicken breast, the absorption intensity at 650 nm of the ring-closing NPs remain to decrease for ~70%. In the revised manuscript, we have also added "It is also found that the RClosed NPs can effectively change to ROpen NPs even if the 610 nm red light irradiation (0.3 W cm^{-2}) is through a 1 cm thickness of chicken breast (Supplementary Fig. 22)." at line 18-20, page 10 to describe the data.

Responses to the Comments and Suggestions of Reviewer #3

Comments:

The manuscript by Qi et al. reported a new kind of light-driven transformable agent for photoacoustic, fluorescence imaging and photodynamic therapy by turning the energy balance of

photophysics under external light stimuli. The molecular structure, photophysical properties and photoreversibility of the organic molecules were studied. In vitro fluorescence, photoacoustic and ROS generation were investigated, which matched well with the design principle. The closed-ring molecule with good conjugation showed NIR absorption and no radiative decay, thus it could be used for in vivo tumor detection. After surgical resection of the tumors, the molecule changed to open-ring form under red light irradiation, and became a sensitive aggregation-induced emission fluorescent agent to inspect the surgery outcomes. For the residual tumors after debulking surgery, the fluorescent agent could also be used for photodynamic therapy (PDT) to improve the survival rate. The investigation is interesting and important for biomedical imaging/therapy community. The manuscript is well organized and the overall quality is good. Therefore, this manuscript can be published in Nature Communications after addressing the following issues.

Q1. The authors have compared the fluorescence and photoacoustic properties of the prepared organic agent with a semiconducting polymer and organic dye (MB). The authors should provide absorption spectrum of the semiconducting polymer and MB in the supporting information, as they may also have some influence on the photoacoustic performance.

Response: We sincerely thank the reviewer for your careful reading and positive feedback to our work. According to the reviewer's suggestion, we have measured the absorption spectra of the semiconducting polymer nanoparticles (SPNs) and methylene blue (MB), and added them in the revised manuscript as new Supplementary Fig. 24. In the revised manuscript, we have added "The absorption spectra of SPNs, MB, and RClosed NPs in water suggest that they share similar maximal absorption wavelength (Supplementary Fig. 24)." at line 23, page 12 and line 1, page 13.

Q2. It seems that the photoreversibility property of the nanoparticles is slower than the solution. Could the authors give some explanation about this?

Response: In the dilute solution, the UV or red light can irradiate onto nearly all the molecules, and the molecules can efficiently transform to the ring-closed/open form. However, in the

nanoparticle (NP) form, the molecules aggregate within NPs, and the photoreaction process of the molecules is not as sufficient as the solution state. As a result, the photoreversibility property of the NPs is a little slower than the soluble state.

Q3. The ^{13}C NMR of compound DTE-2Br (2) is a little strange, so I advise the authors to check and measure it again. The authors should add the basic characterization (^1H -NMR and HRMS) of RClosed-DTE-TPECM in the supporting information.

Response: According to the reviewer's suggestion, we have checked and measured the ^{13}C NMR of compound DTE-2Br (2), and added it in the revised manuscript (new Supplementary Figure 8). We have also carried out the characterization (^1H NMR and HRMS) of RClosed-DTE-TPECM, and added them in the revised manuscript as new Supplementary Figs 15 and 16. Moreover, "**RClosed-DTE-TPECM.** ^1H NMR (400 MHz, CD_2Cl_2): δ 7.45-7.33 (m, 8H), 7.32-7.26 (m, 2H), 7.26-7.15 (m, 16H), 7.14-7.03 (m, 12H), 2.60 (d, 6H), 1.97 (d, 6H). HRMS (MALDI-TOF, m/z): $[\text{M}]^+$ calcd. for $\text{C}_{77}\text{H}_{50}\text{N}_4\text{S}_2\text{F}_6$, 1208.3381; found, 1208.3350." has also been added in the Methods section in the revised Supplementary Information.

Q4. The authors said "In the ring-closing form, Förster resonance energy transfer (FRET) from TPECM to closed-ring DTE". Could the authors show some spectral data to explain the FRET phenomenon?

Response: According to the mechanism of FRET, when the energy transfers from the donor emitter to the acceptor emitter, there should be changes in the emission spectra of both the donor and acceptor. In our manuscript, for the ring-closing molecule (RClosed-DTE-TPECM), we have suggested that FRET occurs from the fluorescent TPECM donors to the non-emissive ring-closing DTE core (acceptor). This is because several literatures have reported that the ring-closing DTE often acts as a non-emissive acceptor to quench the donor emission by FRET process (*Chem. Rev.* **114**, 12174–12277 (2014); *Adv. Mater.* **25**, 378–399 (2013)). According to the reviewer's comment, we have carefully measured the emission property of RClosed-DTE-TPECM. It is verified that the yellow donor (TPECM) fluorescence is almost vanished, meeting the mechanism of energy transfer. Furthermore, there is no emission of RClosed-DTE-TPECM

detected in either the solution state or nanoparticle form even extending the wavelength to 1200 nm (see Supplementary Fig. 19 in the revised Supplementary Information), indicating negligible increase in acceptor (ring-closing DTE) emission. In a word, the FRET mechanism regarding to ring-closing DTE is suggested according to the literatures; however, the ring-closing DTE itself is non-emissive, thus leading to no elevation of acceptor emission.

Q5. The authors claimed that “there is no detectable photoluminescence (PL) emission of the ring-closing isomer in both THF solution and aggregation form”. I suggest the authors to measure the PL spectra of the ring-closed compound and NPs in the NIR region (up to 1000 nm).

Response: According to the reviewer’s suggestion, we have measured the PL property of both RClosed-DTE-TPECM and RClosed NPs in the NIR spectral region. The result has been added as new Supplementary Fig. 19 in the revised manuscript, which indicates that there is no emission of RClosed-DTE-TPECM detected in either the solution state or nanoparticle form even extending the wavelength to 1200 nm.

Supplementary Figure 19. The photoluminescence property of the RClosed-DTE-TPECM in THF solution and RClosed NPs in aqueous media.

Q6. The authors should provide the absorption spectrum of the open-form molecule in the supporting information.

Response: According to the reviewer's suggestion, we have measured the absorption spectrum of the open-form molecule, and added it in the revised Supplementary Information (new Supplementary Fig. 17). The result shows that there is no absorption in the NIR region of the open-form molecule, revealing the good photoswitchable property of the molecule.

Q7. The authors use red light (610 nm) to change RClosed NPs to ROpen NPs, so would the red-light irradiation cause any thermal effect? I suggest the authors to measure the photothermal effect by using the red-light irradiation as mentioned in the manuscript. In addition, the authors should add another MTT data (RClosed NPs+610 nm light) in Supplementary Figure 23.

Response: We thank the reviewer for this constructive suggestion. We have firstly investigated whether the red light irradiation under the experimental condition of our manuscript (610 nm, 0.3 W cm⁻², 5 min) would lead to thermal effect or not. In this regard, the ring-closed NP suspension was irradiated with 610 nm red light (0.3 W cm⁻²) for 10 min and the temperature of the suspension was monitored by an IR thermal camera (Fluke Shanghai Inc.). The result has been added as new Supplementary Fig. 31 in the revised Supplementary Information, which shows that negligible temperature rise of the ring-closed NP suspension is observed during 10 min of the exposure to red light, suggesting that 610 nm red light (0.3 W cm⁻²) irradiation for 10 min does not cause any photothermal effects.

According to the reviewer's suggestion, the MTT assay of "RClosed NPs+610 nm light" has been studied. In this experiment, the 4T1 cancer cells were incubated with a series of doses of RClosed-YSA NPs for 4 h, followed by washing with 1 × PBS and exposure to 610 nm red light (0.3 W cm⁻²) irradiation for 5 min. After 24 h of culture in fresh medium, 100 μL of freshly prepared MTT solution (0.5 mg mL⁻¹) in culture medium was added into each well. After incubation for 3 h, the supernatant was discarded and the precipitate was dissolved in 100 μL of DMSO with gentle shaking. The absorbance of MTT at 570 nm was measured by the microplate reader (GENios Tecan). Cell viability was expressed by the ratio of the absorbance of the cells incubated with the NPs to that of the cells incubated with culture medium only.

The result has been added as new Fig. 5d in the revised manuscript, which reveals that the treatment of "RClosed-YSA NPs+610 nm light" has negligible cytotoxicity, as evidenced by the

cell viabilities > 90% at all of the tested concentrations, implying the good biocompatibility of the NPs and the harmless of red light irradiation under the experimental condition.

In the revised manuscript, we have added "The aforementioned treatment of "RClosed-YSA NPs + red light irradiation" results in negligible cytotoxicity (Fig. 5d), implying the good biocompatibility of the NPs and the harmless of red light irradiation under the experimental condition. It is noted that 610 nm red light (0.3 W cm^{-2}) irradiation for 10 min does not cause the photothermal effect, as indicated by the negligible temperature rise of RClosed-YSA NP suspension during the exposure to red light (Supplementary Fig. 31)." at line 15-21, page 17.

Q8. In the manuscript, the authors used white light for PDT study, so what's the emission spectrum of the light source?

Response: According to the reviewer's suggestion. We have measured the emission spectrum of the white light source used for PDT study, which is as follows. In the revised manuscript, we have added "white light (400-700 nm)" at line 22, page 14.

Figure R3-1. The emission spectrum of the white light source used for PDT study.

Q9. How about the in vivo biocompatibility of the nanoparticles? Could the authors perform some measurements, such as serum biochemistry assay and complete blood count of the mice?

Response: According to the reviewer's suggestion, we have investigated the *in vivo* biocompatibility of our ring-closing NPs and ring-opening NPs. In this experiment, healthy

BALB/c mice were randomly assigned into 3 groups with each group containing 4 mice. Two groups of mice were intravenously injected with 100 μ L of RClosed-YSA NPs and ROpen-YSA NPs (1.6 mM based on DTE-TPECM molecule), respectively. The mice in the third group were untreated as a control. On day 7 post-injection, all the mice in 3 groups were sacrificed and the blood was collected through cardiac puncture at time of sacrifice for blood chemistry analyses by Tianjin Medical University Cancer Institute and Hospital. Furthermore, on day 7, the normal organs of mice in 3 groups including liver and spleen were excised for histology observation.

The results have been added as new Supplementary Figs 33-35 in the revised Supplementary Information, which indicate that there are no differences between the NP-treated and untreated groups in terms of blood biochemistry data and histological analysis of normal organs, demonstrating the good *in vivo* biocompatibility of our NPs.

Q10. Citation for DFT calculation should be added in the Reference part. I advise the authors to add some references related to fluorescence and/or PA imaging guided PDT, e.g., Nat. Commun., 2014, 5, 4596; Adv. Sci., 2017, 4 (10), 1700085.

Response: According to the reviewer's suggestion, we have added the data for DFT calculation in the revised Supplementary Information (new Supplementary Tables 3 and 4). In addition, "Nat. Commun. 5, 4596 (2014)" and "Adv. Sci. 4, 1700085 (2017)" have been cited as refs 40 and 41 in the revised manuscript.

Supplementary Table 3. Cartesian coordinates of optimized ROpen-DTE-TPECM calculated by the DFT, B3LYP/6-31G(d), Gaussian 09 program.

Atom	X	Y	Z
C	7.1205	-1.0762	-1.5541
C	8.2259	-0.3164	-1.5767
C	8.4519	0.3083	-0.4084
S	7.4262	-0.1276	0.5493
C	6.5939	-0.9691	-0.3206
C	6.6965	-1.8252	-2.5905
C	5.4601	-2.3107	-2.822
C	4.2744	-1.7333	-2.5371
C	4.0068	-0.4324	-2.3135
S	2.5702	-0.1912	-2.1337

C	2.0496	-1.5547	-2.2897
C	3.0912	-2.3676	-2.5362
C	0.7396	-1.8893	-2.2271
C	9.4522	1.1722	-0.1203
C	5.5697	-3.5841	-3.6478
C	7.0509	-3.8203	-3.6744
C	7.6641	-2.4585	-3.5755
F	4.8989	-4.6312	-3.1512
F	5.0678	-3.3309	-4.865
F	8.95	-2.4896	-3.205
F	7.5787	-1.7895	-4.7333
F	7.4566	-4.5086	-4.725
F	7.3955	-4.5167	-2.6022
C	5.4476	-1.7562	0.2709
C	4.935	0.7539	-2.4496
C	-0.2514	-0.9753	-2.1559
C	-1.5498	-1.3144	-2.1115
C	-1.9525	-2.5989	-2.0778
C	-0.9666	-3.5154	-2.1318
C	0.3272	-3.1724	-2.2401
C	9.722	1.6018	1.1289
C	10.7143	2.465	1.3982
C	11.4815	3.0284	0.4425
C	11.2135	2.5841	-0.8026
C	10.2666	1.6696	-1.0706
C	12.4111	3.9869	0.7139
C	12.3271	4.7344	1.854
C	13.4314	4.2427	-0.1502
C	11.1136	5.0772	2.3627
C	13.4694	5.1046	2.4956
C	-3.2562	-2.9578	-1.9471
C	-3.5415	-4.0625	-1.2015
C	-4.2458	-2.2203	-2.5171
C	-2.8148	-4.2803	-0.0718
C	-4.5306	-4.91	-1.5908
C	-4.0308	-1.4557	-3.6099
C	-5.0054	-0.7356	-4.1876
C	-6.247	-0.7581	-3.6854
C	-6.4946	-1.5104	-2.605
C	-5.5092	-2.2256	-2.0392
C	-2.5827	-5.4992	0.4504
C	-1.8478	-5.6855	1.5597
C	-1.2639	-4.6817	2.2491
C	-1.5097	-3.4656	1.7242
C	-2.2477	-3.2724	0.6194
C	-5.2292	-5.6631	-0.7137

C	-6.2113	-6.4929	-1.0992
C	-6.5335	-6.5977	-2.3951
C	-5.868	-5.8552	-3.2897
C	-4.8919	-5.0258	-2.8876
C	13.5514	6.2366	3.2285
C	14.6812	6.6149	3.8471
C	15.7861	5.8637	3.7514
C	15.7405	4.7366	3.0292
C	14.6028	4.3731	2.4163
C	13.9522	5.4838	-0.2729
C	14.9608	5.7626	-1.1136
C	15.4972	4.793	-1.866
C	15.0132	3.5486	-1.7585
C	14.0027	3.2869	-0.9144
C	10.0146	5.221	1.5981
C	8.801	5.48	2.1136
C	8.5591	5.6159	3.433
C	9.6756	5.5009	4.1845
C	10.8935	5.2436	3.6806
C	-0.5189	-4.9248	3.3623
C	0.2788	-4.0289	3.992
C	-0.5592	-6.3358	3.9534
C	7.3431	5.8362	4.0077
C	6.1512	5.8808	3.3642
C	7.3009	6.0344	5.5268
C	0.9738	-4.3682	5.0627
N	1.5909	-4.6605	5.9996
C	0.5466	-2.7864	3.639
N	0.8259	-1.6983	3.3505
C	5.0256	6.0898	4.0237
N	4.0343	6.2774	4.5951
C	5.9	5.714	2.0801
N	5.6341	5.5748	0.9596
H	8.8421	-0.2042	-2.4781
H	2.9948	-3.4424	-2.7248
H	5.256	-2.7041	-0.2768
H	4.5145	-1.1494	0.291
H	5.6506	-2.0747	1.3192
H	4.4386	1.6232	-2.9389
H	5.2882	1.1074	-1.4554
H	5.8075	0.5397	-3.1032
H	-0.0445	0.1082	-2.1359
H	-2.2789	-0.4908	-2.0163
H	-1.2005	-4.5946	-2.1292
H	1.0386	-4.0112	-2.2949
H	9.1547	1.2375	2.0025

H	10.8719	2.6896	2.4664
H	11.7211	3.0077	-1.6853
H	10.1313	1.4141	-2.1338
H	-3.0491	-1.4285	-4.1121
H	-4.7912	-0.1356	-5.0883
H	-7.0522	-0.1705	-4.1562
H	-7.511	-1.5315	-2.1765
H	-5.7813	-2.791	-1.1325
H	-2.9415	-6.4101	-0.059
H	-1.712	-6.7411	1.8447
H	-1.1662	-2.5388	2.1986
H	-2.4173	-2.2206	0.3303
H	-5.0624	-5.5883	0.3738
H	-6.769	-7.0777	-0.3479
H	-7.3393	-7.2768	-2.718
H	-6.1212	-5.9341	-4.3606
H	-4.3683	-4.4703	-3.6838
H	12.7015	6.9361	3.3049
H	14.71	7.5567	4.4209
H	16.7181	6.1704	4.254
H	16.6403	4.1036	2.9477
H	14.6337	3.4137	1.8729
H	13.5369	6.3408	0.2847
H	15.345	6.7936	-1.1956
H	16.3281	5.0143	-2.5562
H	15.4626	2.7356	-2.3537
H	13.6987	2.2293	-0.8337
H	10.0851	5.1458	0.4986
H	8.0156	5.5765	1.3564
H	9.6532	5.5636	5.2839
H	11.7022	5.0874	4.4157
H	-1.6051	-6.7048	4.0549
H	-0.1689	-6.4259	4.9884
H	0.0392	-7.042	3.3334
H	7.5894	5.0965	6.0549
H	6.3154	6.3077	5.9559
H	7.9603	6.8756	5.8403

Supplementary Table 4. Cartesian coordinates of optimized Closed-DTE-TPECM calculated by the DFT, B3LYP/6-31G(d), Gaussian 09 program.

Atom	X	Y	Z
------	---	---	---

C	3.5903	1.7062	-2.8518
C	2.2587	1.5284	-2.8077
C	1.8257	0.2478	-2.7953
S	2.9628	-0.6793	-2.9558
C	4.3471	0.4381	-3.1609
C	4.2807	2.8332	-2.6022
C	5.6052	2.7941	-2.3465
C	6.2548	1.6293	-2.1741
C	5.4603	0.3465	-2.1221
S	6.8554	-0.7257	-2.4534
C	7.9549	0.1161	-1.9334
C	7.5593	1.4089	-1.931
C	9.1485	-0.3862	-1.541
C	0.5696	-0.1558	-2.4905
C	6.1324	4.1868	-2.0998
C	4.8762	4.8295	-1.5802
C	3.7562	4.2342	-2.3875
F	7.1693	4.2793	-1.2603
F	6.5087	4.7059	-3.2761
F	2.5576	4.3284	-1.8024
F	3.6583	4.8137	-3.5921
F	4.9102	6.1481	-1.6231
F	4.7203	4.4894	-0.3102
C	4.8186	0.451	-4.6258
C	4.9702	0.1245	-0.6738
C	9.4867	-1.6841	-1.693
C	10.6215	-2.2027	-1.1989
C	11.4781	-1.4627	-0.4694
C	11.1803	-0.1518	-0.3763
C	10.0711	0.3753	-0.9203
C	0.2454	-1.4363	-2.2191
C	-0.9598	-1.7948	-1.7453
C	-1.9556	-0.9187	-1.4929
C	-1.6828	0.3118	-1.9772
C	-0.4697	0.6945	-2.4071
C	-3.1534	-1.262	-0.935
C	-4.1401	-0.3808	-0.5963
C	-3.4046	-2.5812	-0.7486
C	-3.9267	0.8969	-0.1766
C	-5.3785	-0.8376	-0.3013
C	12.5127	-2.0296	0.2051
C	12.8005	-1.5553	1.452
C	13.1747	-3.088	-0.3335
C	11.7851	-1.0654	2.215
C	14.0736	-1.5693	1.9281
C	13.2692	-3.2554	-1.6709

C	13.9302	-4.2848	-2.2234
C	14.5267	-5.1964	-1.4441
C	14.4505	-5.0601	-0.1135
C	13.7881	-4.0231	0.4236
C	10.5199	-1.5178	2.1219
C	9.5025	-0.9875	2.8187
C	9.6403	0.0422	3.6817
C	10.9169	0.462	3.7925
C	11.9424	-0.0631	3.1001
C	14.3427	-1.6632	3.2489
C	15.596	-1.7129	3.727
C	16.6383	-1.6678	2.8871
C	16.4056	-1.574	1.5712
C	15.1462	-1.5287	1.1083
C	-6.4158	-0.641	-1.1373
C	-7.6368	-1.1122	-0.8387
C	-7.8374	-1.7897	0.3016
C	-6.8127	-1.9923	1.1429
C	-5.5938	-1.5186	0.8415
C	-4.4128	-3.2226	-1.3723
C	-4.6633	-4.5218	-1.1461
C	-3.9103	-5.2118	-0.2774
C	-2.9087	-4.591	0.3617
C	-2.6642	-3.2924	0.1267
C	-4.8794	1.8484	-0.1299
C	-4.6262	3.1164	0.2365
C	-3.4106	3.5589	0.6232
C	-2.5003	2.5696	0.7043
C	-2.7493	1.3015	0.3399
C	8.5618	0.5616	4.3337
C	8.5783	1.5804	5.2274
C	7.1896	-0.0487	4.0306
C	-3.1868	4.8702	0.9146
C	-1.9754	5.4339	1.1417
C	-4.3951	5.8095	0.9462
C	9.6089	2.277	5.6652
N	10.4858	2.9131	6.0797
C	-1.8577	6.7257	1.3916
N	-1.7466	7.8598	1.606
C	7.4683	2.0097	5.8014
N	6.4998	2.3908	6.3127
C	-0.7877	4.8616	1.0917
N	0.2727	4.3939	1.0423
H	1.5981	2.3824	-2.6174
H	8.2453	2.245	-1.747
H	5.5729	1.2486	-4.8103

H	5.2789	-0.52	-4.9163
H	3.9772	0.6386	-5.3317
H	4.4988	-0.8759	-0.5478
H	4.2239	0.8911	-0.3663
H	5.7963	0.1806	0.0717
H	8.8207	-2.4086	-2.1913
H	10.7512	-3.2941	-1.3022
H	11.8473	0.5354	0.1736
H	9.9152	1.4512	-0.7438
H	0.9817	-2.2537	-2.3074
H	-1.059	-2.872	-1.5419
H	-2.4552	1.0948	-2.0336
H	-0.3743	1.7513	-2.7045
H	12.8452	-2.5192	-2.3753
H	13.9975	-4.3757	-3.3208
H	15.0719	-6.0442	-1.8907
H	14.9271	-5.8117	0.5386
H	13.7357	-3.9991	1.5247
H	10.2714	-2.3769	1.4752
H	8.5315	-1.4702	2.6258
H	11.2072	1.2976	4.4383
H	12.9259	0.4219	3.2296
H	13.5383	-1.7677	3.9967
H	15.7717	-1.8141	4.8115
H	17.6702	-1.7144	3.2724
H	17.2556	-1.5307	0.8697
H	15.0319	-1.4233	0.0164
H	-6.2693	-0.1045	-2.0907
H	-8.4757	-0.95	-1.5358
H	-8.8389	-2.1809	0.5458
H	-6.973	-2.5484	2.0819
H	-4.7693	-1.6934	1.5543
H	-5.0397	-2.7021	-2.1153
H	-5.4865	-5.0261	-1.6795
H	-4.1157	-6.2782	-0.0867
H	-2.2953	-5.1468	1.0909
H	-1.8561	-2.807	0.7023
H	-5.9112	1.6471	-0.4583
H	-5.4916	3.7931	0.1522
H	-1.4935	2.7285	1.1075
H	-1.9314	0.58	0.5108
H	6.3266	0.4143	4.5505
H	6.9388	0.0571	2.9499
H	7.1526	-1.1191	4.3378
H	-5.1893	5.4206	1.6234
H	-4.803	5.9599	-0.0799

H	-4.2026	6.8288	1.3399
---	---------	--------	--------

Reviewers' Comments:

Reviewer #1:

Remarks to the Author:

The authors made remarkable improvement for the revised manuscript. I have no further comment and recommend an acceptance by the Journal.

Reviewer #2:

Remarks to the Author:

The authors have addressed my concerns. The new experiments further validate the probe. The scavenging experiments and mouse experiments now show further demonstrate the utility of the probe. The authors should double-check that the information about Forester distance is in the manuscript. Other than that minor point, I recommend publication in the journal.

Reviewer #3:

Remarks to the Author:

The authors have successfully addressed all the issues raised. This manuscript can be accepted and published as it is.

Responses to the Comments and Suggestions of Reviewer #1

Comments:

The authors made remarkable improvement for the revised manuscript. I have no further comment and recommend an acceptance by the Journal.

Response: We sincerely thank the reviewer for your careful reading and recommendation for publication of our work.

Responses to the Comments and Suggestions of Reviewer #2

Comments:

The authors have addressed my concerns. The new experiments further validate the probe. The scavenging experiments and mouse experiments now show further demonstrate the utility of the probe. The authors should double-check that the information about Förster distance is in the manuscript. Other than that minor point, I recommend publication in the journal.

Response: We sincerely thank the reviewer for your careful reading and recommendation for publication of our work. According to the non-fluorescent closed-ring dithienylethene (DTE) structures in the literatures (Giordano, L., Jovin, T. M., Irie, M. & Jares-Erijman, E. A. *J. Am. Chem. Soc.* **124**, 7481 (2002). Li, C. *et al. Nat. Commun.* **5**, 5709 (2014)), we explain the non-fluorescence property of RClosed-DTE-TPECM by “intramolecular energy transfer from TPECM to closed-ring DTE”. That’s why we haven’t discussed about “Förster distance” in the manuscript.

Responses to the Comments and Suggestions of Reviewer #3

Comments:

The authors have successfully addressed all the issues raised. This manuscript can be accepted and published as it is.

Response: We sincerely thank the reviewer for your careful reading and recommendation for publication of our work.